# HIDDEN NO MORE: ATTACKING AND DEFENDING PRIVATE THIRD-PARTY LLM INFERENCE

**Rahul Thomas**[1,3], **Louai Zahran**[1], **Erica Choi**[1,2], **Akilesh Potti**[1], **Micah Goldblum**[1,2], **Arka Pal**[1*]

[1]Ritual
[2]Columbia University
[3]Stanford University

## ABSTRACT

Recent advances in Large Language Models (LLMs) have led to widespread adoption of third-party inference services, raising critical privacy concerns. In this work, we introduce a novel reconstruction technique that can recover original prompts from hidden states with nearly perfect accuracy across multiple state-of-the-art LLMs in the increasingly important open-weights setting. Although the attack is conceptually simple, it has not – to the best of our knowledge – previously been described nor shown to work practically. Furthermore, our attack remains effective against various permutation and noise-based defenses, challenging assumptions about the security of previously proposed schemes. To address these vulnerabilities, we propose Cascade, a multi-party inference scheme that leverages sharding in the sequence dimension to retain privacy of the user input. Through theoretical analysis and empirical evaluation, we demonstrate that Cascade is secure against both our attack as well as previous methods, while maintaining computational and communication efficiency. Our findings highlight the importance of rigorous security analysis in privacy-preserving LLM inference and offer practical solutions for secure deployment.

## 1 INTRODUCTION

Modern large language models (LLMs) now often comprise hundreds of billions of parameters, necessitating significant hardware resources for deploying them for inference. In particular, recent *open-weights* models demonstrate cutting-edge performance (DeepSeek-AI et al., 2025; Qwen et al., 2025), but remain difficult for many to run. Individuals and organizations have therefore begun to increasingly rely on third-party LLM inference services that host these models. This raises significant privacy implications, particularly in domains where confidentiality of data is paramount, such as healthcare, finance and legal applications, and in jurisdictions where data privacy is subject to regulations (e.g. GDPR in Europe). As such, a growing area of research interest is the creation of inference methodologies and schemes that protect the privacy of user prompts.

One approach to privacy-preserving-inference is based on having multiple parties participate jointly in performing the inference, with the idea that each party cannot itself reconstruct the input solely with the information that it is given in the protocol. This approach is known as Secure Multi-Party Computation (SMPC) and has a long history of application to general functions (Yao, 1982; Goldreich et al., 1987). Recently, the methodologies of SMPC have been applied to LLMs (Huang et al., 2022; Hao et al., 2022; Pang et al., 2023; Akimoto et al., 2023; Dong et al., 2023a; Li et al., 2024). However, SMPC methods introduce significant computational and communication overhead, particularly so at non-linearities in the model.

Therefore, other works seek to mitigate the punitive costs of standard SMPC approaches by additionally utilizing statistical obfuscation approaches. In particular, recent work (Zheng et al., 2024; Yuan et al., 2024; Luo et al., 2024) has leveraged the permutation-equivariance properties of transformers (Xu et al., 2024) to propose permutation-based schemes for private inference. Under these schemes, hidden states are revealed as permuted plaintext to the party performing the inference. These works justify security by referring to the extremely large set of possibilities in the permutation space, and concluding that the reversal of these permuted states to the original user prompts is practically infeasible.

---

*Corresponding author: `arka@ritual.net`.

Table 1: Percentage of perfectly decoded evaluation samples under our **vocab-matching attack**, at different layers of Gemma-2-2B-IT and Llama-3.1-8B-Instruct.

| Layer | Gemma | Llama |
|-------|-------|-------|
| 1 | 100% | 100% |
| 6 | 100% | 100% |
| 11 | 100% | 100% |
| 16 | 100% | 100% |
| 21 | 100% | 99.9% |
| 26 | 100% | 99.7% |

In this paper, we show that the above schemes are not secure. We devise a new method of attack that is capable of nearly perfect decoding of the user input in the open-weights setting, improving on existing work (Wan et al., 2024). We further show that this attack maintains nearly perfect decoding performance under a variety of permutation types, including those relied upon by the schemes above. Furthermore, the attack is capable of decoding against common noising methods proposed in the literature for private inference (Morris et al., 2023a).

We then introduce a new multi-party scheme, **Cascade**, that is resistant to our attack, that leverages *sharding* at the token level instead of permutations or noise for obfuscation. We show that Cascade is also resistant to existing reversal approaches in the literature (Wan et al., 2024; Morris et al., 2023b). While Cascade does not provide the rigorous privacy guarantees of cryptographic MPC schemes, it is much more efficient than them, and presents a new paradigm in the trade-off between scalability and security.

## 2 SETUP & THREAT MODEL

We assume the setting of a user $U$ who wishes to perform inference with an LLM model $M$ on some input prompt $x$, which can be considered as an ordered sequence of tokens $[x_1, x_2, ..., x_N]$. We denote the size of the hidden state of the LLM by $d$, and the sequence length by $N$.

As the user $U$ does not have the resources to perform the inference themselves, they rely on a set of third-parties $P_1, P_2, ..., P_K$. The model weights of $M$, including the embedding lookup table, are considered to be known to all parties; although, as we discuss later in Section 5 and Appendix C, this assumption can be relaxed. We consider the setting where each of the parties behaves *semi-honestly*, a common assumption of past works (Zheng et al., 2024; Luo et al., 2024; Dong et al., 2023a; Yuan et al., 2024). Semi-honest parties will follow the defined protocol faithfully, but may exploit any information that they receive during the execution of the protocol to attempt to recover the user's data.

## 3 RELATED WORK

Several existing works have investigated the reversibility of LLM embeddings into the original sentence inputs (Song & Raghunathan, 2020; Morris et al., 2023a; Li et al., 2023b; Kugler et al., 2024) with relatively good decoding performance. Different from our setting, these focus on reversal of a single vector $e = \phi(x) \in \mathbb{R}^d$, where $\phi$ is an embedding model that returns a single fixed-size vector from an $N$-token input $x = [x_1, x_2, ..., x_N]$. In our paper, we are instead concerned with the reversibility of full intermediate states $[h_1, h_2, \ldots, h_N] \in \mathbb{R}^{N \times d}$ of an LLM.

The closest two previous works on reversibility in our setting are those of Wan et al. (2024) and Morris et al. (2023b). The former work focuses on reversal of hidden states in general, whilst the latter is particularly focused on logit output distribution reversal. In both papers, the authors use a learnt transformer-based network to reverse the sequence of hidden states into the original token inputs. Experiments are conducted on two decoder-based models, Llama-2-7B and ChatGLM-6B. Average F1 scores of approximately $60\%$ are achieved across a range of datasets in Wan et al. (2024) on hidden states near the last layers of the models, and scores around $75\%$ are achieved for logit reversal in Morris et al. (2023b). Importantly, the latter paper does not assume any access by the adversary to model weights, whilst the former explicitly denotes the case of a model provider performing inference on user provided embeddings, and so is more analogous to our setting.

Petrov et al. (2024) propose an attack that shares some elements with ours below – especially, exploitation of the unidirectional nature of decoder-based LLMs, as well as the finite and discrete space of LLMs' vocabularies. However, they are concerned primarily with the setting of gradient reversal into original inputs in the federated *training* setting – different from our focus on private *inference*. Furthermore, their method relies on full-rank properties of the gradients, which are not always satisfied (e.g. when the prompts are longer than the hidden dimension size). By contrast, our method does not have any such restrictions.

## 4 HIDDEN STATE REVERSAL

We begin by considering the general case where one of the parties performing inference, $P_k$, receives an intermediate sequence of hidden states $\boldsymbol{h} = [h_1, h_2, ..., h_N]$ at some layer $l$ of the LLM $M$.

Can the party $P_k$ reverse the hidden states $\boldsymbol{h}$ to the input sequence of tokens $\boldsymbol{x} = [x_1, x_2, ..., x_N]$ that produced $\boldsymbol{h}$?

### 4.1 VOCABULARY-MATCHING ATTACK

Our proposed decoding scheme in the above setting leverages the causal ordering of decoder-only transformers, as well as the finite set of possibilities of input tokens.

The attack begins with a batched forward pass over all length-1 sequences $[v]$, where $v$ ranges over words in the vocabulary $\mathcal{V}$. From this, the adversary gets $V = |\mathcal{V}|$ candidate layer $l$ hidden states $\boldsymbol{h}(v) \in \mathbb{R}^{1 \times d}$. They set the first predicted input token $\widehat{x}_1$ to be the token $v$ for which $\boldsymbol{h}(v)$ exactly matches the first hidden state $h_1$.

Next, the adversary performs a batched forward pass over all length-2 sequences $[\widehat{x}_1, v]$ with $v \in \mathcal{V}$, to get $V$ candidate layer $l$ hidden states $\boldsymbol{h}(\widehat{x}_1, v) \in \mathbb{R}^{2 \times d}$. Now, they set the second predicted input token $\widehat{x}_2$ to be the token $v$ where the second row of $\boldsymbol{h}(\widehat{x}_1, v)$ equals the second hidden state $h_2$.

In general, at the $n$th stage, using the first $n-1$ predicted input tokens $\widehat{x}_1, \dots, \widehat{x}_{n-1}$, the adversary performs a forward pass over all length-$n$ sequences $[\widehat{x}_1, \dots, \widehat{x}_{n-1}, v]$ with $v \in \mathcal{V}$. They obtain $V$ candidate layer $l$ hidden states $\boldsymbol{h}(\widehat{x}_1, \dots, \widehat{x}_{n-1}, v) \in \mathbb{R}^{n \times d}$, and set the $n$th predicted input token $\widehat{x}_n$ to be the token $v$ where the $n$th (last) row of candidate states matches the $n$th hidden state $h_n$. Iterating over $n = 1, \dots, N$, the adversary sequentially obtains the predicted input sequence $\widehat{\boldsymbol{x}}$ from the layer $l$ hidden states $\boldsymbol{h}$.

Although naively one may expect that an exact match of $\boldsymbol{h}$ would require exponential search (specifically, over all $V^N$ possible sequences of tokens $\boldsymbol{x}$), we see that by exploiting the autoregressive property of transformers, this is reduced to a linear search; the total cost of this attack is $O(VN)$.

### 4.2 PRACTICAL IMPLEMENTATION

#### 4.2.1 NON-DETERMINISM

Although the above attack is conceptually simple, there are two important implicit assumptions. The first is that there is only one – or at worst, a small number – of matches that are found at each step. If the average number of matches at each step is $M$, then the search space grows approximately as $M^N$, which is infeasible when $M$ or $N$ is large. Prior work (Dong et al., 2023b) has demonstrated the rank-reducing effects of attention blocks, so it is plausible that the size of the subspace in latter layers in particular is too small to prevent large numbers of collisions.

Secondly, and more subtly, there is an assumption that the forward pass performed over the vocabulary will match the forward pass that generated the given hiddens $\boldsymbol{h}$ exactly. In general, due to the non-associativity of floating-point operations (Villa et al., 2009) this will not be the case. Particularly in the GPU setting with parallel asynchronous thread execution and pooling without global synchronization, there can be considerable variation in the output (Shanmugavelu et al., 2024). In addition, differences in hardware, random number seeds, environment variables and the state of initialized memory on the machine can all add to the variability, and these values may not be known to the adversary.

Due to the presence of this reducible and irreducible noise, we find that exact matching cannot be used successfully with this attack. Thus, we loosen our matching requirements by computing the *L1-distance* between the last row of candidate hidden states and the given hidden state, and accept a match for a token $v$ if the distance is below some threshold $\epsilon$. If no such match is found, we choose the token $v$ which gives minimal L1-distance.

---

**Algorithm 1** Vocabulary-Matching Attack

---

**input** Model $M$, layer $l$ hidden states $\boldsymbol{h} = [h_1, \ldots, h_N]$, vocabulary $\mathcal{V}$, proposal model $P$, L1-threshold $\epsilon$
**output** Decoded token sequence $\widehat{\boldsymbol{x}} = [\widehat{x}_1, \widehat{x}_2, \ldots, \widehat{x}_N]$
 1: Initialize empty sequence $\widehat{\boldsymbol{x}} \leftarrow []$
 2: **for** $i = 1$ to $N$ **do**
 3:    $\mathcal{V}_{\text{ordered}} \leftarrow \text{argsort}(P(\widehat{\boldsymbol{x}}))$ {Get ordered vocabulary from proposal model}
 4:    min_dist $\leftarrow \infty$
 5:    best_match $\leftarrow$ None
 6:    **for** $v \in \mathcal{V}_{\text{ordered}}$ **do**
 7:       $g \leftarrow M_{\leq l}([\widehat{\boldsymbol{x}}, v])$ {Forward pass up to layer $l$}
 8:       dist $\leftarrow \|g - h_i\|_1$ {Calculate L1 distance}
 9:       **if** dist $<$ min_dist **then**
10:          min_dist $\leftarrow$ dist
11:          best_match $\leftarrow v$
12:       **end if**
13:       **if** dist $< \epsilon$ **then**
14:          $\widehat{x}_i \leftarrow v$
15:          break
16:       **end if**
17:    **end for**
18:    **if** dist $\geq \epsilon$ **then**
19:       $\widehat{x}_i \leftarrow$ best_match
20:    **end if**
21: **end for**
22: **return** $\widehat{\boldsymbol{x}}$

---

However, by allowing an $\epsilon$-ball for matching, we increase the possibility of collisions as stated above. Is our attack still successful – i.e., are LLM states sufficiently non-colliding – even with this fuzzy matching? In Section 4.3, we find the answer is emphatically yes.

### 4.2.2 EFFICIENCY

We optimize runtime in practice using a *proposal model* to provide a likelihood-based order of iteration through the vocabulary. We find that this modification reduces the average number of tokens searched through at each step from $V/2$ to $\sim 100$, resulting in a speedup of more than $1000\times$. Further, we implement a novel variation of key-value-caching (KV-caching) to reduce the computational cost of our attack. Further details on these optimizations are given in Appendix A. With these efficiency improvements, we reduce the decoding time of prompts of length 50 from many hours to typically less than 30 seconds. Our final algorithm is outlined in Algorithm 1.

### 4.3 EXPERIMENTS & DISCUSSION

We conduct our experiments on two state-of-the-art open-source LLMs, Gemma-2-2B-IT (Team et al., 2024) and Llama-3.1-8B-Instruct (Grattafiori et al., 2024). These models have different sizes (numbers of parameters), training methodologies, and architectures. We conduct testing on samples from the Fineweb-Edu dataset (Penedo et al., 2024). We test on every fifth layer in the targeted models, extracting the hidden states of layers 1, 6, 11, 16, 21, and 26.

For each layer of interest, we tune $\epsilon$ by performing a ternary search on a small training set comprising 50 prompts taken from FineWeb, to determine the optimal L1-threshold under which predicted tokens are accepted as matches. We evaluate on 1000 held out prompts, and our results are shown in Table 1. We find that nearly all evaluation samples are perfectly decoded. Accompanying $\epsilon$ values are given in Appendix B. Due to computational constraints, each evaluation prompt was truncated to a maximum of 50 tokens; however, small-scale experiments with prompts exceeding 200 tokens demonstrated that our results generalize to longer prompt settings – vocab-matching still perfectly decodes all hidden states into their corresponding tokens. The success of our attack also allows us to conclude that LLM hidden states are highly distinct and non-colliding.

## 5 PERMUTED HIDDEN STATE REVERSAL

We now consider the case where one of the parties performing inference receives a permutation of the intermediate sequence of hidden states $h$ at some layer $l$ of the LLM $M$.

### 5.1 EXISTING WORK

Recently, a number of works have proposed utilizing permutations to perform privacy-preserving inference in a multi-party-computation (MPC) setup.

Zheng et al. (2024) permute at the non-linear components of the LLMs in order to reveal them 'safely' to one of the parties, and therefore avoid expensive iterated inter-party communication. The permutation is done on the attention logits before the softmax, at layer normalizations, and at the non-linear functions in the MLP block. The latter is a purely elementwise function, so the authors can do a full permutation across the $[N, d]$ elements, resulting in a permutation space of size $(Nd)!$. However, softmax and layer-norm are row-wise operations, so the permutation applied in this case is a (distinct) permutation to the columns, followed by a permutation of the $N$ rows, resulting in a permutation space of size $N!(d!)^N$.

Yuan et al. (2024) permute both the model weights and the user prompt embeddings in the hidden $d$ dimension, and the entire inference process (on the next token) is then carried out by a single party.

Luo et al. (2024) applies ideas from both the above works. The proposed method permutes the model weights, utilizing additive secret-sharing for the linear layers, but relies on two-party permuted plaintext computation at the non-linearities (softmax, layer-norm and GeLU). Permutation is applied in the hidden $d$ dimension.

### 5.2 PERMUTED INTERMEDIATE STATES ARE NOT SAFE

We now propose a modification of our vocab-matching attack introduced in Section 4, which breaks user input privacy for the above schemes in the open-weights setting. Extensions to the attack also break privacy in the closed-weights setting that Yuan et al. (2024) and Luo et al. (2024) originally consider: see Appendix C for details. Luo et al. (2024) discuss theoretical considerations for why permutations should be statistically secure – we discuss why these considerations do not anticipate or mitigate our attack in Appendix D.

#### 5.2.1 SEQUENCE-DIM PERMUTATION

Assume that sequence permutation has been applied to layer $l$ hidden states $h = [h_1, h_2, ..., h_N]$:

$$h_{\text{seq\_perm}} = [h_{\sigma(1)}, h_{\sigma(2)}, ..., h_{\sigma(N)}],$$

where $\sigma$ is a permutation of $[N] = \{1, 2, \ldots, N\}$. Then, we modify the vocab-matching attack as follows. At the $n$th stage, we now choose the vocabulary token $v$ where the $n$th row of the corresponding candidate hidden state is within an L1-distance of $\epsilon$ from *any* row of $h_{\text{seq\_perm}}$. Suppose this $\epsilon$-ball match is made with the $i$th row $h_{\sigma(i)}$. We set the $n$th predicted input token $\widehat{x}_n$ to $v$, and remove $h_{\sigma(i)}$ from consideration for hidden state matching in *all* future stages. Iterating over $n = 1, \ldots, N$, we obtain the predicted input sequence $\widehat{x}$ from sequence-permuted hidden states $h_{\text{seq\_perm}}$.

Compared to the vocab-matching attack, the opportunities for collision are now increased up to $N$-fold, as we match with up to $N$ rows of $h$ rather than one. However, we again observe very few collisions in practice and are able to decode the vast majority of input prompts: see Table 2.

#### 5.2.2 HIDDEN-DIM PERMUTATION

Next we consider the case where permutation has been performed on the hidden dimension of $h$ instead. That is, the party performing inference is now given:

$$h_{\text{hidden\_perm}} = [\pi_1(h_1), \pi_2(h_2), ..., \pi_N(h_N)]$$

where each $\pi_i$ permutes elements of a $d$-dimensional vector. In this setting, it is no longer possible to use L1-distance directly to find the nearest vocabulary token match. We instead use the **sorted L1-distance**, which individually sorts the two vectors to be compared and then computes their L1-distance. Again, the existence of noise may appear to be a significant obstacle to this approach. However, we find that even this relatively simple matching approach is robust enough to noise to achieve nearly perfect decoding. Our results are shown in Table 2.

Table 2: The percentage of evaluation samples that were perfectly decoded under sequence-dim, hidden-dim, and factorized 2D permutations, for Gemma-2-2B-IT and Llama-3.1-8B-Instruct.

| Layer | Sequence-Dim | | Hidden-Dim | | Factorized-2D | |
|---|---|---|---|---|---|---|
| | Gemma | Llama | Gemma | Llama | Gemma | Llama |
| 1 | 100% | 99.7% | 100% | 100% | 99.9% | 98.4% |
| 6 | 99.8% | 100% | 100% | 98.5% | 99.5% | 97.8% |
| 11 | 100% | 100% | 100% | 99.2% | 99.5% | 98.9% |
| 16 | 100% | 100% | 99.9% | 99.4% | 99.2% | 98.8% |
| 21 | 99.8% | 100% | 98.2% | 98.9% | 99.1% | 98.0% |
| 26 | 99.8% | 100% | 98.0% | 98.2% | 99.0% | 97.6% |

### 5.2.3 FACTORIZED-2D PERMUTATION

We now consider the case of a factorized two-dimensional permutation as used in Zheng et al. (2024), where a hidden-dimension permutation is applied to each hidden state, and then these resulting states are shuffled in the sequence dimension. The adversary now has:

$$\boldsymbol{h}_{\text{fact\_perm}} = [\pi_1(h_{\sigma(1)}), \pi_2(h_{\sigma(2)}), ..., \pi_N(h_{\sigma(N)})]$$

where $\sigma$ is a permutation of $[N]$ and each $\pi_i$ permutes a $d$-dimensional vector. The attack in this setting again utilizes the sorted-L1 matching function, but now expands to consider all $N$ rows of $\boldsymbol{h}_{\text{fact\_perm}}$, as in Section 5.2.1. Remarkably, even in this setting, the hidden states of both models are decoded nearly perfectly across layers (Table 2).

We conclude that permuted hidden states of LLMs are highly decodeable by our attack, and therefore schemes which expose them are not secure in the open-weights setting.

## 6 NOISED & QUANTIZED HIDDEN STATE REVERSAL

Section 5 shows that modifications to our attack can successfully decode any sequence-dimension, hidden-dimension, and factorized-2D permutation of the hidden states. We now examine the efficacy of our attack on alternative methods of defense that modify the hidden states directly – such as by adding noise, or by quantizing the model to a lower precision. We find that generally, these methods are still not sufficient to defend against our attack. Due to space constraints, we provide much further detail in Appendix E.

## 7 CASCADE: TOKEN-SHARDED MULTI-PARTY INFERENCE

As Section 5 and Section 6 show permutations and noising of hidden states are not secure, a natural follow-up question is whether *sharded* hidden states are secure. We suggest the answer is affirmative for certain sharding schemes. We propose a defense to the vocab-matching attack based on *token-dimension sharding* of hidden states, which leads to a new multi-party inference scheme: **Cascade**.

Notably, Cascade does not use cryptographic primitives; the actual computations are nearly unchanged from a standard forward pass, so almost no additional computational overhead is incurred. There is also no degradation of performance, as no approximations need to be made (as is typical for SMPC schemes in non-linearities). The scheme also does not require any user (or trusted party) interaction during inference.

### 7.1 SCHEME DESCRIPTION

At a high level, Cascade exploits the fact that only the self-attention mechanism in transformers has interaction between the tokens in a sequence; for all other parts of the architecture, the tokens are treated similarly to batch dimension elements.

**Sharding** In multi-headed attention, we denote $H$ as the attention heads, $H_{KV}$ as key-value heads (if grouped-query attention is used), $N$ as the token count, $d_{emb}$ as the hidden dimension, and $d$ as the attention hidden dimension. There are three axes of sharding used along the token dimension: **(1)** the sharding of hidden states $\boldsymbol{h} \in \mathbb{R}^{N \times d_{emb}}$, **(2)** the sharding of query states $\boldsymbol{q} \in \mathbb{R}^{H \times N \times d}$, and **(3)** the sharding of key and value states $\boldsymbol{k}, \boldsymbol{v} \in \mathbb{R}^{H_{KV} \times N \times d}$. These involve splitting token indices $[N] = \{1, 2, \ldots, N\}$ into a union of disjoint

subsets $\{R_i\}_{i=1}^{\alpha}$ for hiddens, $\{S_j\}_{j=1}^{\beta}$ for queries, and $\{T_k\}_{k=1}^{\gamma}$ for keys and values, where $\alpha, \beta, \gamma$ are shard counts. These index shardings will also be used on the positional embeddings $\boldsymbol{p} \in \mathbb{R}^{N \times d_{emb}}$ and attention mask $\boldsymbol{m} \in \mathbb{R}^{N \times N}$, which are initialized pre-inference. We use the following shorthand notation for sharded states:

$$\boldsymbol{h}_i^R = \boldsymbol{h}[R_i] \quad \boldsymbol{p}_i^R = \boldsymbol{p}[R_i] \quad \boldsymbol{m}_{jk}^{ST} = \boldsymbol{m}[S_j, T_k]$$
$$\boldsymbol{q}_i^R = \boldsymbol{q}[:, R_i] \quad \boldsymbol{q}_j^S = \boldsymbol{q}[:, S_j] \quad \boldsymbol{q}_{ij}^{RS} = \boldsymbol{q}[:, R_i \cap S_j]$$
$$\boldsymbol{k}_i^R = \boldsymbol{k}[:, R_i] \quad \boldsymbol{k}_k^T = \boldsymbol{k}[:, T_k] \quad \boldsymbol{k}_{ik}^{RT} = \boldsymbol{k}[:, R_i \cap T_k]$$

We define $\boldsymbol{v}_*^*$ from $\boldsymbol{v}$ in the same way that $\boldsymbol{k}_*^*$ is defined from $\boldsymbol{k}$. Denoting the masked attention logits by $\boldsymbol{a} \in \mathbb{R}^{H \times N \times N}$, we also define $\boldsymbol{a}_{ik}^{RT} = \boldsymbol{a}[:, R_i, T_k]$, $\boldsymbol{a}_{jk}^{ST} = \boldsymbol{a}[:, S_j, T_k]$, and $\boldsymbol{a}_{ijk}^{RST} = \boldsymbol{a}[:, R_i \cap S_j, T_k]$, as well as the associated quantities $\boldsymbol{e}_{*k}^{*T} = \text{expsum}(\boldsymbol{a}_{*k}^{*T})$ and $\boldsymbol{u}_{*k}^{*T} = \text{softmax}(\boldsymbol{a}_{*k}^{*T})\boldsymbol{v}_k^T$. In Cascade, sharding of all of these matrices aims to prevent each node from reversing tokens.

**Nodes**  There are two types of nodes which hold the sharded states above: **CompNodes** and **AttnNodes**. We initialize $\alpha$ CompNodes and $\beta^2$ AttnNodes, indexed as CompNode$_i$ and AttnNode$_{jk}$ for all $i \in [\alpha]$ and $j, k \in [\beta]$.

**Inference**  Cascade breaks down single layer inference into the **pre-pass by CompNodes**, **attention-pass by AttnNodes**, and **post-pass by CompNodes**. So per layer, there is CompNode to AttnNode communication after the pre-pass and AttnNode to CompNode communication after the attention-pass, and both are parallelizable. We provide a detailed outline of these procedures in Appendix F: see Algorithm 2 for a high-level overview of a single layer pass. Cascade single layer pipelines can be stacked because each CompNode$_i$ starts and ends with access to only token indices $R_i$. After the last layer, CompNodes apply the LM head to get $R_i$-sharded logits, and the CompNode with access to the last token will use this generate the next token[1]. This token's embedding is sent back to the CompNode holding the next token index in $R_i$ to append to its layer 0 hidden states, and Cascade inference repeats to generate the next token. In practice, next-token generation can be sped up with KV-caching, which is compatible with Cascade: see Appendix N for optimization details.

**Symmetrization**  To simplify security and cost analysis, we force our scheme to be *symmetric* with respect to query and key-value sharding. That is, we set $\gamma = \beta$ and each $S_j = T_j$. In Appendix I, we show that any non-symmetric scheme can be made symmetric at no loss of security.

## 7.2 SECURITY ANALYSIS

In this section, we examine the security properties of Cascade. Cascade does not employ cryptographic techniques with provable guarantees of security, so our analysis can only elucidate on statistical security. Nevertheless, we examine a diverse range of security considerations below.

The security of Cascade is a function of its implementation parameters: the *number of nodes* participating, as well as the *sharding strategy* used, i.e. $\{R_i\}_{i=1}^{\alpha}$ and $\{S_j\}_{j=1}^{\beta}$.

As the space of sharding strategies is vast, we focus on a particular class of strategies called $(\boldsymbol{c}, \boldsymbol{\delta})$-**sharding**, where each sharded index set takes the form of a 'clustered-arithmetic' or $(c, \delta)$-sequence. We focus on these strategies as they fulfill several security desiderata, which we discuss below. Nevertheless, we emphasize that other strategies may be preferable to $(c, \delta)$-sharding depending on the use-case.

**Definition 7.1.** We say a subset of indices $[N]$ is a $(c, \delta)$-sequence if it takes the form

$$\{i, i+1, \ldots, i+c-1, \delta+i, \delta+i+1, \ldots, \delta+i+c-1, 2\delta+i,$$
$$2\delta+i+1, \ldots, 2\delta+i+c-1, \ldots\}$$

for some $i$. That is, the token indices follow an arithmetic progression given by the $\delta$ parameter, but with 'clusters' of length $c$.

$(c, \delta)$ **vs. Number of CompNodes**  Under $(c, \delta)$-sharding, the minimum number of CompNodes $\alpha$ needed to ensure all indices in $[N]$ are held by some node is given by $\alpha = \lceil \delta/c \rceil$.

---

[1]The logits can also be returned to the user if desired.

Table 3: ROUGE scores of text reconstruction from the hiddens of layer 1 of Gemma-2-2B-IT for various values of $c$ and $\alpha$ under $(c, \delta)$-sharding. Increasing $c$ or $\alpha$ results in worse reconstruction.

|        | $\alpha = 4$ | $\alpha = 8$ | $\alpha = 12$ |
|--------|--------------|--------------|---------------|
| $c = 1$ | 0.701 | 0.467 | 0.349 |
| $c = 4$ | 0.427 | 0.290 | 0.230 |
| $c = 8$ | 0.355 | 0.222 | 0.191 |

### 7.2.1 COMPNODE SECURITY

There are two questions to be asked regarding security – first, can we select a sharding strategy that defends against our attack outlined in Section 4? And second – do we remain secure to learning based attacks as in prior works Wan et al. (2024) and Morris et al. (2023b)?

**Vocab-Matching Attack**  Our main line of defense against vocab-matching is to ensure large token gaps in nodes. For a $(c, \delta)$-sharding scheme, in each shard, the distance from one 'cluster' of indices to the next is $\delta - c + 1$. Therefore, in order to carry out the attack, we cannot perform a single run through $\mathcal{V}$ and obtain the next token match; we must search over length $\delta - c + 1$ sequences of infilled words. This scales exponentially: the work done is now $V^{\delta-c+1}$. Given that $V \sim O(100000)$ in typical modern LLMs[2], this is likely already infeasible in practice for $\delta - c + 1 \geq \rho$ when the *vocab matching threshold* $\rho$ is set to 3, say. Note $\rho$ may be set to suit the security demands of the use case. In general, preventing brute-force vocab-matching is equivalent to $(\alpha - 1)c \geq \rho - 1$, so we can increase $\alpha$ or $c$ to maintain security. For $\rho = 3$, all $\alpha, c \geq 2$ satisfy this.

**Learning-Based Attacks**  We now consider learning-based reversal attacks. We conduct experiments on Gemma-2-2B-IT and Llama-3.1-8B-Instruct. We fine-tune both models on random $(c, \delta)$-masked input sequences from FineWeb-Edu, with the target labels being the full input sequence of tokens. We update both models to use a bidirectional attention-mask, in line with the token-infilling nature of this task. We train until the eval loss on a held-out set converges, over representations on layer 1 from each model; evaluation is performed on the same layer's hidden states. Our approach is similar to previous works Wan et al. (2024); Morris et al. (2023b).

Our results for Gemma-2-2B-IT are shown in Table 3. We see that when $c, \alpha \geq 8$, we have a ROUGE score less than 0.25, showing significant reconstruction difficulty. We further tested on $c = 4, \alpha = 16$ and $c = 8, \alpha = 24$ and obtained ROUGE scores of 0.1733 and 0.1439, supporting that security continues to improve by scaling $c$ and $\alpha$.
We further examine (a) the choice of hidden layer, and (b) whether Llama representations are better decoded, in Appendix G. We find similar or better security across these other parameter choices.

**Layer 0**  Layer 0 of the LLM is a special case, as the hidden states at this layer are just the token embeddings, which are immediately reversible to their original tokens. We conclude that **Cascade should not be used when the security of every token is paramount**. It is only applicable for scenarios where the revelation of at least some subset of the input is considered acceptable. [3] We examine further elements of security at Layer 0 in Appendix H.

### 7.2.2 ATTNNODE SECURITY

So far, we analyzed the security of CompNodes in isolation. We examine the security of AttnNodes via $S$-sharding in Appendix I. Finally, we examine other sources of information leakage and restrictions on sharding schemes in Appendix M.

### 7.3 PERFORMANCE

We now examine the performance of Cascade. First, we provide theoretical estimates of computational and communication costs associated with Cascade. As in previous SMPC works (Li et al., 2023a; Dong et al.,

---

[2]One may argue that the distribution of the text makes the base of the exponential much lower than $V$. However, the work is still exponential, so large enough $c$ and $\alpha$ still prevents vocab-matching.

[3]If individual token security is strongly necessary, existing SMPC protocols could be used for Layer 0 alone, at the cost of additional computational and communication overhead. Detailed investigation of this idea is left to future work.

2023a; Li et al., 2024), we assume perfect parallel transport in communication, a homogeneous (outgoing) node-wise bandwidth of $B$, and an inter-node latency of $\tau$. We also denote $F$ as the number of bytes per element. Then, we perform experiments to substantiate our estimates, and further, investigate the total runtime of Cascade for inference on BERT and BERT-Large.

### 7.3.1 THEORETICAL ESTIMATES OF COMPUTATION AND COMMUNICATION COSTS

**Computation**   In terms of floating point operations, there is little overhead from Cascade compared to vanilla inference: see Appendix J for a detailed analysis. We provide empirical evidence for this below in Section 7.3.2.

**Communication**   We present single layer communication byte and time overhead formulae. Full derivations are given in Appendix K:

$$\text{CommBytes} = \beta F(2dH + 2dH_{KV} + 2H) \cdot N \tag{1}$$

$$\text{CommTime} = 2\tau + \frac{\beta F d(H + 2H_{KV})}{B} \cdot \max_i |R_i| + \frac{F(d+2)H}{B} \cdot \max_j |S_j|. \tag{2}$$

In Equation (1), only $\beta$ depends on the sharding scheme. Therefore, byte overhead is minimized when $\beta$ is minimized, i.e. with minimal AttnNodes. This overhead is independent of sharding for fixed $\beta$, and scales linearly with $\beta$. Furthermore, communication time is minimized when both $\beta \max_i |R_i|$ and $\max_j |S_j|$ are minimized. For fixed $\alpha, \beta$, since $\{R_i\}_{i=1}^{\alpha}$ and $\{S_j\}_{j=1}^{\beta}$ partition $[N]$, then $\max_i |R_i| \geq \lceil N/\alpha \rceil$ and $\max_j |S_j| \geq \lceil N/\beta \rceil$. Equality is achieved when all $R_i$, and all $S_j$, are around the same size. Thus, for fixed CompNode and AttnNode counts, $(c, \delta)$-sharding achieves optimal communication time, as it has nearly uniform shard sizes.

### 7.3.2 PERFORMANCE EXPERIMENTS

We now run experiments to evaluate the real-world performance of Cascade. We compare against two recent SMPC schemes for LLM inference, MPCFormer (Li et al., 2023a) and Puma (Dong et al., 2023a). We reimplement these in their original frameworks of CrypTen (Knott et al., 2021) and SPU (Ma et al., 2023) respectively. In both cases, for fair comparison, we set the model weights to be public, whilst maintaining the inputs as private.

We implement Cascade using the distributed computing framework Ray (Moritz et al., 2018). Our experiments are conducted on Paperspace machines with 16 vCPU and 64GB RAM, with the CPU model being Intel Xeon Gold 6226R CPU @ 2.90GHz. All machines are colocated in the same region with an average measured bandwidth of 2 Gbps and an average measured latency of 0.38 ms. We perform inference (a single forward pass) on Bert-Base and Bert-Large with an input prompt of 128 tokens – we repeat this measurement 100 times as there is run-to-run variability in the timing.

Our results are shown in Table 4 below. The data for MPCFormer and Puma is taken from Dong et al. (2023a).

Table 4: Total runtime means and 95% confidence intervals for a single forward pass on Bert-Base and Bert-Large for MPCFormer, Puma, and Cascade with different settings, compared to standard (vanilla) inference. Runtimes are given in seconds. A prompt length of 128 is used. Values for MPCFormer and Puma are based on reimplementations in CrypTen (Knott et al., 2021) and SPU (Ma et al., 2023) to reflect the open-weights setting.

| Scheme | Bert-Base Runtime (s) | Bert-Large Runtime (s) |
|---|---|---|
| MPCFormer | 49.401 | 143.883 |
| Puma | 28.356 | 61.957 |
| Cascade$_{\alpha=1}$ (no Ray) | 0.109 [0.100, 0.134] | 0.320 [0.231, 1.067] |
| Cascade$_{\alpha=1}$ | 0.319 [0.306, 0.355] | 1.012 [0.967, 1.089] |
| Cascade$_{\alpha=4}$ | 0.588 [0.513, 0.688] | 1.572 [1.441, 1.734] |
| Cascade$_{\alpha=8}$ | 0.742 [0.622, 0.962] | 1.584 [1.271, 1.965] |
| *Vanilla* | 0.091 [0.084, 0.121] | 0.273 [0.200, 0.993] |

We begin by measuring the performance of Cascade with $\alpha = 1$ and without using Ray, to obtain a direct comparison of the computational cost of Cascade vs. vanilla inference without any Ray overhead or inter-node

communication. As we can see from Table 4, the mean runtime is 109ms vs. 91ms for vanilla inference for Bert-Base, and 320ms vs. 273ms for Bert-Large. Profiling reveals this minor increase is entirely explained by the attention-score compilation step discussed in Appendix J. Nevertheless, the mean runtime remains within the 95% confidence interval of vanilla inference in both cases.

We next show the performance of Cascade with $\alpha = 1$ and using Ray. As seen above, this is slower than not using Ray by around a factor of $3\times$. This slowdown can be attributed to framework-specific overhead, such as serialization and deserialization introduced by Ray in order to send and receive objects. In other words, Cascade is so efficient that the distributed-compute framework overhead now constitutes a significant proportion of its slowdown from vanilla inference, rather than protocol-specific overhead.

Finally, we measure the performance of Cascade with $\alpha = 4, 8$, which are values that can be used in real applications to maintain privacy, as discussed in Section 7.2. Consequently, the experimental setup involves cluster configurations of 6 and 18 distinct machines for $\alpha = 4$ and 8 respectively. We see that performance in these cases is slowed relative to the $\alpha = 1$ case, due to the introduction of communication overhead between the machines. However, Cascade remains $\sim 90\times$ faster than MPCFormer for Bert-Large ($\sim 65\times$ faster for Bert-Base) and $\sim 40\times$ faster than Puma for both models even in its slowest tested $\alpha = 8$ setting.

Furthermore, although it is difficult to make strong conclusions from 3 data points, we observe that the increase in runtime for both models from $\alpha = 1$ to $\alpha = 8$ appears sublinear, even as the number of distinct machines increases superlinearly.

To support our theoretical estimates in Section 7.3.1, we also examined the total communication in the $\alpha = 1$ case using tshark. We found that the total communicated bytes are within 2% of Equation (1). The extra bytes can be attributed to Ray-specific metadata and a slight increase from the raw bytes size due to serialization. We present a comparative table of total communicated bytes for Cascade versus MPCFormer and Puma in Table 5. We see that even in the most expensive $\alpha = 8$ setting, Cascade is $\sim 160\times$ more efficient in total bytes transferred than MPCFormer, and $\sim 140\times$ more efficient than Puma.

Table 5: Total gigabytes (GB) communicated for a single forward pass on Bert-Base and Bert-Large for MPCFormer, Puma, and Cascade with different settings, compared to standard (vanilla) inference. A prompt length of 128 is used. Values for MPCFormer and Puma are taken from Dong et al. (2023a).

| Scheme | Bert-Base (GB Communicated) | Bert-Large (GB Communicated) |
|---|---|---|
| MPCFormer | 12.089 | 32.577 |
| Puma | 10.773 | 27.246 |
| Cascade$_{\alpha=1}$ | 0.009 | 0.025 |
| Cascade$_{\alpha=4}$ | 0.038 | 0.101 |
| Cascade$_{\alpha=8}$ | 0.076 | 0.203 |

We conclude that Cascade is significantly faster and more communication-efficient than prior SMPC methods, and offers a new paradigm in the trade-off between scalability and security for private LLM inference.

## 8 CONCLUSION & FUTURE WORK

We have identified a new attack for decoding LLM hidden states into their original user text in the increasingly important open-weights setting. This attack obtains near perfect accuracy on decoding even permuted hidden states, effectively invalidating the security of some existing MPC schemes. We also introduced a novel multi-party scheme, Cascade, that uses token sharding to defend against our attack, as well as existing attack methods in the literature. Future directions of work could further examine the space of sharding strategies, particularly in unreliable network settings.

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

## A    VOCAB-MATCHING ATTACK OPTIMIZATIONS

Although the cost of the attack outlined in Section 4 is linear in $V$, the size of vocabularies can be quite large in practice. For example, Gemma-2-2B-IT has a vocabulary size of 256000. Therefore we seek to optimize this by introducing a *proposal model*. The purpose of the proposal model is to provide a suggested ordering over the vocabulary, rather than iterate through it in an arbitrary order. It does so by taking in the token sequence that has been partially decoded so far and producing the next-token logits. We then search through the next-token logits in decreasing order of probability. In practice, we find that this modification reduces the expected number of tokens searched through at each step from $V/2$ to approximately 100, thus representing a constant factor speedup of more than $1000\times$.

Moreover, we implement a novel variation of key-value-caching (KV-caching) to reduce the computational time of our attack. Note that at the $n$th stage of the decoding, we are performing a $V$-batched forward pass on $[\widehat{x}_1, \widehat{x}_2, ..., \widehat{x}_{n-1}, v]$ over $v \in \mathcal{V}$, where $\widehat{x}_1, \widehat{x}_2, \ldots, \widehat{x}_{n-1}$ are the tokens that we have already decoded. As this forward pass needs to be repeated many times for different $v$ but the same $\widehat{x}_i$, we cache the keys and values associated to the $\widehat{x}_i$ and reuse them across all forward passes. This is different from standard KV-caching, which stores the keys and values for generation over a single sequence: here, we reuse keys and values across many sequences.

## B    OPTIMAL $\epsilon$ FOR DECODING

We report the full set of optimal $\epsilon$ thresholds in decoding, for each permutation type below. We observe that generally, the optimal $\epsilon$ increases in later layers across all permutation types – which may be due to the effect of the reducible and irreducible noise we mention in Section 4 taking up a larger subspace volume as it propagates to deeper layers. We also observe that Llama tends to have much lower $\epsilon$ values in general. Further investigation of these interesting trends and their implications for the properties and structure of LLM hidden states is left to future work.

Table 6: Optimal $\epsilon$ thresholds for hidden state reversal with no permutation, over various Gemma-2-2B-IT and Llama-3.1-8B-Instruct layers.

| Layer | Gemma | Llama |
|-------|-------|-------|
| 1 | 22.0 | 0.6 |
| 6 | 70.0 | 7.1 |
| 11 | 204.0 | 18.3 |
| 16 | 293.0 | 29.0 |
| 21 | 400.0 | 76.0 |
| 26 | 318.0 | 156.0 |

Table 7: Optimal $\epsilon$ thresholds for hidden state reversal with sequence dimension permutation, over various Gemma-2-2B-IT and Llama-3.1-8B-Instruct layers.

| Layer | Gemma | Llama |
|-------|-------|-------|
| 1 | 12.8 | 1.4 |
| 6 | 72.6 | 3.3 |
| 11 | 229.0 | 7.4 |
| 16 | 301.0 | 7.4 |
| 21 | 385.0 | 26.6 |
| 26 | 220.0 | 29.6 |

Table 8: Optimal $\epsilon$ thresholds for hidden state reversal with hidden dimension permutation, over various Gemma-2-2B-IT and Llama-3.1-8B-Instruct layers.

| Layer | Gemma | Llama |
|-------|-------|-------|
| 1     | 12.5  | 0.5   |
| 6     | 25.0  | 3.5   |
| 11    | 45.0  | 3.7   |
| 16    | 73.0  | 5.2   |
| 21    | 118.0 | 6.3   |
| 26    | 61.0  | 9.8   |

Table 9: Optimal $\epsilon$ thresholds for hidden state reversal with factorized-2D permutation, over various Gemma-2-2B-IT and Llama-3.1-8B-Instruct layers.

| Layer | Gemma | Llama |
|-------|-------|-------|
| 1     | 21.0  | 0.3   |
| 6     | 26.0  | 3.0   |
| 11    | 47.0  | 9.0   |
| 16    | 69.0  | 9.0   |
| 21    | 118.0 | 14.0  |
| 26    | 51.0  | 14.0  |

## C  BREAKING PERMUTATION SCHEMES IN THE CLOSED-WEIGHTS SETTING

In Section 5, we showed that permuted hidden states are highly reversible. This means that permutation-based privacy-preserving schemes like PermLLM (Zheng et al., 2024), Secure Transformer Inference Protocol (Yuan et al., 2024) and Centaur (Luo et al., 2024) are not secure in the *open-weights* setting. The distinction from closed-weights is important here: to carry out the vocab-matching attack to decode permuted hidden states at layer $l$, the attacker must have access to all model parameters at layers $\leq l$, as well as the embedding lookup table.

Because of this, we cannot immediately claim to break the security of Centaur or the Secure Transformer Inference Protocol (STIP), which explicitly tackle the closed-weights setting where the inference provider *does not* have access to model weights. Nevertheless, for both of these schemes, we show that slight modifications can be made to the vocab-matching attack to break security even in this setting.

**STIP**     There are three parties: the model developer $P_1$, the model server $P_2$ (who carries out inference), and the user $P_3$. The goal of STIP is to have $P_2$ carry out inference on $P_3$'s input, protect $P_1$'s private model weights $\Theta$ from $P_2$ and $P_3$, and protect $P_3$'s private input data from $P_1$ and $P_2$. This is accomplished with random permutation in the hidden dimension. At initialization, $P_1$ sends random $d \times d$ permutation matrices $\pi, \pi_c$ to the user $P_3$, where $d$ is the token embedding dimension. They also randomly permute each weight matrix or vector in the row and/or column dimensions, to obtain the altered model weights $\Theta'$; these are given to the model server $P_2$, who cannot recover $\Theta$ from them. Then during inference, instead of sending their private input data $X \in \mathbb{R}^{N \times d}$, the user encrypts it with permutation $\pi$, i.e. they send $X\pi$. Afterwards, a standard transformer forward pass is carried out, but with the weights $\Theta$ (unknown to the model server $P_2$) replaced by permuted weights $\Theta'$. Finally, the results are sent to the user, who applies permutation $\pi_c$ to obtain the output of the inference. The STIP authors show through orthogonality of permutation matrices that the final output is the true output of inference.

Because STIP involves a standard transformer forward pass with permuted weights $\Theta'$, it follows that the developer $P_2$ sees permutations of certain intermediate states. For instance, at Layer $0$, after applying the altered $Q, K, V$-projections on token embeddings, they have hidden dimension permutations of the true $Q, K, V$-projections. In fact, at all layers $l$, the layer $l$ altered hidden states are hidden dimension permutations of the true layer $l$ hidden states. This means the attack described in Section 5, with forward passes from the *altered transformer model with weights* $\Theta'$, could be applied to decode the hidden states – but only if the vocabulary of input embeddings is known, which is not the case in the closed-weights setting. However, there are two possible ways to sidestep this issue. Firstly, for many models – such as Gemma – the embedding

matrix is simply the transpose of the language-modeling head, whose permutation is known to $P_2$. Therefore, the vocabulary embedding vectors to search over in this case are simply the (permuted) rows of the transpose of this matrix – and these permutations can be uncovered by matching against the (permuted) input embedding vectors received from the use to start inference. Secondly, even if the language-modeling head is not the same as the embedding matrix, $P_2$ can still collect the vocabulary over repeated observations of input prompts given – there will necessarily be a finite set of these.

The final step to decoding would then be the mapping of the decoded embeddings into tokens. If the tokenizer is not publicly revealed, this may seem difficult at first – but note that this essentially constitutes a simple substitution cipher, where each token in the vocabulary is substituted by its embedding vector. Again, by collecting data over many queries and using simple methods such as frequency analysis and positional information, $P_2$ can learn to decode this into the original tokens; substitution ciphers are in general easily broken given sufficient data.

**Centaur**    Centaur followed the three-party threat model of STIP, and attempted to reconcile two problems. On the model weight privacy side, they aimed to prevent exposure of the lookup table to the user. On the user privacy side, they wanted to avoid exposing certain unpermuted intermediate results (like the matrices $QK^T$ at each layer, due to the $Q$ and $K$ permutations cancelling). To do this, they introduced *additive secret sharing* between the developer $P_1$ and server $P_2$ at most stages of self-attention, only requiring reconstruction of additive shares (by the developer) during nonlinearities. Although this resolves the previous two concerns, it is still the case that permutations of true layer $l$ hidden states are exposed to the model developer at nonlinearities, and that the developer has access to permuted (in the same way as in STIP) weights $\Theta'$ and a permuted lookup table. Furthermore, a forward pass through the first $l$ layers of a transformer model with weights $\Theta'$ would still yield hidden dimension permutations of a forward pass through $l$ layers with weights $\Theta$. Thus, like with STIP, a vocab-matching attack with the permuted model and permuted lookup table can be used to reverse engineer the input text, and frequency analysis and context can break the token substitution cipher.

## D    DISTANCE CORRELATION DOES NOT GUARANTEE PERMUTATION SECURITY

Here, we contextualize statistical arguments on the security of permuted hidden states. In particular, we clarify why they do not anticipate the vocab-matching attack, which emphatically shows that permuted hidden states are *not* secure. One common justification is through statistical security: Zheng et al. (2022) and later Zheng et al. (2024) measure input leakage from permuted hidden states with *distance correlation* (Székely et al., 2007). They theoretically and experimentally show that the expected distance correlation for a random one-dimensional linear projection is larger than that of a random permutation, i.e. one would expect a random permutation to leak less information than one-dimensional compression. After introducing Centaur, Luo et al. (2024) use the same argument to justify the security of their scheme.

There are several reasons why this result cannot be used to make strong guarantees on permutation security:

1. **The result does not capture key ingredients of transformer architectures.**

   First, they consider distance correlation for a particular $1 \times d$ embedding after linear projection. However, for their result to apply to the case of LLM inference, it should instead be proven for the full $N \times d$ embedding matrix – and it should account for other transformer components. Particularly, it should consider self-attention, in which tokens are not processed independently. In fact, the unidirectional nature of decoder-only LLMs (e.g. each token output only depends on previous token inputs) is what enables the vocab-matching attack. Consequently, their distance correlation result, which ignores this dependence, fails to anticipate such an attack.

   Also, they do not use any discrete or combinatorial information, which is available in the form of the lookup embedding table. This is also key to the vocab-matching attack, and would not be captured in the expected distance correlation. The result on distance correlation fails to show *discrete* reversals are difficult. For example, the first token output only depends on the first token input – why search over a full distribution of first token embedding vectors, when we can iterate through the finitely many vectors in the lookup table to reverse the first input token?

2. **Expected distance correlation does not give formal security guarantees.**

First, it is possible to have the distance correlation between random variables $W$ and $X$ be greater that of $Y$ and $Z$, but have the reconstructibility of $W$ from $X$ be harder than that of $Y$ from $Z$. Reconstructibility has many interpretations: here, we define it as the maximum possible probability of reconstructing the input within a given $\delta$-threshold, over all estimators. This reconstruction up to *absolute* error (called $\delta$-reconstruction) is especially relevant for practical implementations of attacks like vocab-matching, where non-determinism forces us to choose a match within a given *absolute* error (see Section 4 for further details).

For an explicit example of the above paradoxical scenario, define independent $W, \varepsilon \sim \mathcal{N}(0, 1)$. Let $Y$ come from an arbitrary symmetric distribution about zero, and construct

$$X = \rho W + \sqrt{1 - \rho^2}\varepsilon, \quad Z = |Y|$$

where $0 < \rho < 1$. Using standard properties of normal random variables, one can see $X \sim \mathcal{N}(0, 1)$, and the correlation between $X$ and $W$ is $\rho$. Thus, by Theorem 7 in Székely et al. (2007), which lower bounds distance correlation of standard normals in terms of (Pearson) correlation, we have $\text{DistCorr}(W, X) > 0.89\rho$. Furthermore, by Theorem 1 in **?**, which upper bounds the distance correlation of a symmetric random variable and its absolute value, we have $\text{DistCorr}(Y, Z) \leq 2^{-1/4}$. Therefore, we have $\text{DistCorr}(W, X) > \text{DistCorr}(Y, Z)$ whenever $\rho > \frac{2^{-1/4}}{0.89} \approx 0.945$. However, even in this case, it turns out that $\delta$-reconstruction of $W$ from $X$ can sometimes be more difficult than $\delta$-reconstruction of $Y$ from $Z$. For instance, it is impossible to reconstruct $W$ from $X$ within a threshold of $\delta = 0.1$ with $100\%$ probability regardless of what estimator is used: to do so would require reconstructing $\varepsilon$ from $X$ within a certain threshold with $100\%$ probability, but this is impossible as it is independent from $X$. But if $Y$ is defined to always lie in $[-0.1, 0.1]$, then one can simply estimate $\widehat{Y} = 0$ and *always* reconstruct $Y$ within an error of $\delta = 0.1$. That is, $\delta$-reconstructibility of $Y$ from $Z$ is easier than $\delta$-reconstructibility of $W$ from $X$ for $\delta = 0.1$, even as $(Y, Z)$ has lower distance correlation than $(W, X)$.

Secondly, the inequality used by Luo et al. (2024) is an expectation over linear projections and permutations – therefore, it is possible that there are some linear weights (and also some permutations) for which the distance correlation of a random permutation is in fact smaller than that of a random 1D linear projection.

3. **Reconstruction from random 1D projections of LLM hidden states is possible.**

A key part of their reasoning is that reconstructing input from a random 1-dimensional linear projection is difficult. However, there is no theoretical reason that this should be the case for such projections of LLM hidden states. Consider the vocab-matching attack, performed with a matching function that matches each hidden state to a vocabulary embedding using a randomly-weighted sum (i.e. a random 1D projection) of its elements. This attack would still successfully reverse the inputs in this case: we have shown experimentally with L1-distance matching that LLM hidden states are in general highly non-colliding, and the below theorem then implies randomly-weighted sums of LLM hidden states are also highly non-colliding. That is, the vocab-matching attack can be used to reconstruct inputs from random 1D linear projections of LLM states.

**Proposition D.1.** *Suppose random weights $\boldsymbol{w} \in \mathbb{R}^d$ are draw from a $d$-variate spherically symmetric distribution $\mathcal{D}$. Then any $\boldsymbol{x}, \boldsymbol{y} \in \mathbb{R}^d$, we have the absolute difference of $\boldsymbol{w}$-weighted sums of $\boldsymbol{x}$ and $\boldsymbol{y}$ exceeds the L1 distance between $\boldsymbol{x}$ and $\boldsymbol{y}$, meaning*

$$\left| \sum_{i=1}^{d} w_i x_i - \sum_{i=1}^{d} w_i y_i \right| \geq \sum_{i=1}^{d} |x_i - y_i|, \tag{3}$$

*with probability $\geq P_{\boldsymbol{\gamma} \sim \mathcal{D}}(|\gamma_1| \geq \sqrt{d})$.*

*Proof.* Denote $\boldsymbol{z} = \boldsymbol{x} - \boldsymbol{y}$. Observe that

$$\left| \sum_{i=1}^{d} w_i x_i - \sum_{i=1}^{d} w_i y_i \right| = \left| \sum_{i=1}^{d} w_i (x_i - y_i) \right| = \left| \sum_{i=1}^{d} w_i z_i \right| = |\boldsymbol{w}^T \boldsymbol{z}|.$$

Thus, Equation (3) is equivalent to $|\boldsymbol{w}^T\boldsymbol{z}| \geq \|\boldsymbol{z}\|_1$. Then, from the standard bound $\|\boldsymbol{z}\|_1 \leq \sqrt{d}\|\boldsymbol{z}\|_2$, which can be proven by an application of Cauchy-Schwarz, we see that Equation (3) holds whenever

$$|\boldsymbol{w}^T\boldsymbol{z}| \geq \sqrt{d}\|\boldsymbol{z}\|_2. \tag{4}$$

We now aim to compute the probability of the above event. Choose a $d \times d$ orthogonal matrix $Q$ such that $\boldsymbol{z}_q := Q\boldsymbol{z} \in \mathbb{R}^d$ only has a nonzero coordinate $L$ in its first position. By orthogonality and the fact that $\mathcal{D}$ is spherically symmetric, we see $\boldsymbol{w}_q := Q\boldsymbol{w}$ has distribution $\mathcal{D}$. Furthermore, orthogonal linear transformations are length-preserving (by L2 norm), so we have $\|\boldsymbol{z}_q\|_2 = \|Q\boldsymbol{z}\|_2 = \|\boldsymbol{z}\|_2 = |L|$. In fact, as $Q^TQ = I$, observe that $\boldsymbol{w}^T\boldsymbol{z} = \boldsymbol{w}^TQ^TQ\boldsymbol{z} = (Q\boldsymbol{w})^T(Q\boldsymbol{z}) = \boldsymbol{w}_q^T\boldsymbol{w}_z$. Hence, Equation (4) becomes

$$|\boldsymbol{w}_q^T\boldsymbol{z}_q| = |L||(\boldsymbol{w}_q)_1| \geq |L|\sqrt{d}.$$

This is equivalent to saying the first coordinate of $\boldsymbol{w}_q$ has magnitude at least $\sqrt{d}$. But we showed $\boldsymbol{w}_q$ has distribution $\mathcal{D}$, so the probability of Equation (4) is precisely $P_{\gamma \sim \mathcal{D}}(|\gamma_1| \geq \sqrt{d})$. This is therefore a lower bound on the probability of Equation (3), since we showed Equation (3) holds whenever Equation (4) does. $\qquad\square$

We can obtain an exact bound above by setting $\mathcal{D}$ as a multivariate Gaussian. That is, for $\boldsymbol{w} = (w_1, \ldots, w_d)$, we i.i.d. sample each $w_i \sim \mathcal{N}(0, \sigma)$. Then $P_{\gamma \sim \mathcal{D}}(|\gamma_1| \geq \sqrt{d}) = P_{\gamma \sim \mathcal{N}(0, \sigma)}(|\gamma| \geq \sqrt{d}) = 2 - 2\Phi(\sqrt{d}/\sigma)$. So, by setting $\sigma$ sufficiently large, we can make $\Phi(\sqrt{d}/\sigma) \to \Phi(0) = \frac{1}{2}$, and the probability lower bound approaches $2 - 2 \cdot \frac{1}{2} = 1$. For instance, for $d = 4096$ (as in Llama-3.1-8B-Instruct), if we sample weights in this manner with standard deviation $\sigma = 256$, the probability lower bound becomes $2 - 2\Phi(0.25) \approx 80\%$.

Essentially, by sampling weights with large enough variance, we can ensure random 1D projections of two vectors are non-colliding with high probability, whenever the vectors are non-colliding by L1 distance. Further work should experimentally verify the efficacy of vocab-matching with randomly-weighted sums, in the presence of non-determinism and other practical implementation considerations.

## E    NOISED & QUANTIZED HIDDEN STATE REVERSAL

Here, we provide full details on our experiments and results highlighted in Section 6. We investigate the following methods of modification to the hidden states of Gemma-2-2B-IT:

1. Diagonal Gaussian noise with mean 0 and standard deviation $\sigma$ applied to each hidden dimension in the input embeddings, as proposed in Morris et al. (2023a).

2. A randomly generated embedding inserted as a prefix to the original sequence. This has the effect of modifying the subsequent hidden states via self-attention. We generate this embedding from a Gaussian with means and standard deviations of each hidden dimension set to the average over the token vocabulary $\mathcal{V}$.

3. Quantization of the model from original 16-bit to 8-bit and 4-bit, using the bitsandbytes library (BitsAndBytes, 2025).

Table 10: ROUGE reconstruction scores across 50 evaluation samples for various noising methods and permutation types. The LiveBench column is a measure of downstream performance, with higher values indicating stronger performance.

| Method | Unpermuted | Sequence | Hidden | LiveBench |
|---|---|---|---|---|
| Baseline (no noise) | 1.00 | 1.00 | 1.00 | 20.7 |
| Gaussian, $\sigma = 10^{-2}$ | 0.84 | 0.78 | 0.04 | 21.0 |
| Gaussian, $\sigma = 10^{-1}$ | 0.81 | 0.33 | 0.01 | 1.2 |
| Random emb. prefix | 0.80 | 0.57 | 0.14 | 21.3 |
| 8-bit quantization | 0.89 | 0.86 | 0.75 | 20.2 |
| 4-bit quantization | 0.88 | 0.84 | 0.75 | 19.1 |

It is clear that with large enough noise, the efficacy of our attack can be blunted. However, doing so may also degrade downstream performance on tasks of interest. To measure the effect of this for each of the above methods, we utilize LiveBench (White et al., 2024), a benchmark that tests multiple different components of LLM peformance such as language, reasoning and math. Furthermore, we tested the effect of combining each of these methods with sequence and hidden dimension permutations. Our results are shown in Table 10.

We see that unpermuted hidden states are still highly decodeable via our attack under all methods tested. Remarkably, even 4-bit quantization is not sufficient to introduce enough collisions to significantly hamper sequence-permuted decoding. Sequence dimension permutation affords some security with high Gaussian noise injection, but this severely hampers downstream performance. The combination of hidden dimension permutation and Gaussian noise with standard deviation $10^{-2}$ appears largely secure, and so offers potential as a solution to the insecurity of the schemes in Yuan et al. (2024) and Luo et al. (2024). Note that LiveBench scores carry some variability – the original paper shows variation of around $\pm 2$ for the scores is within a 95% bootstrapped confidence interval – and so the baseline, Gaussian with standard deviation $10^{-2}$, and random embedding prefix methods are all within noise in performance.

A full breakdown of the LiveBench scores by category and the ROUGE scores by layer of each of the above methods and permutation types is given in Appendix O.

## F    CASCADE SCHEME DETAILS

We provide further details on the pre-pass, attention-pass and attention pass components of Cascade, as introduced in Section 7.1. Furthermore, the full algorithm of Cascade is given in Algorithm 2. All notation is defined in Section 7.1. Below, all shard-specific slicing and concatenation operations are initialized in the node setup. Furthermore, we assume that all AttnNodes wait until all CompNodes finish the pre-pass to do the attention-pass, and all CompNodes wait until all AttnNodes finish the attention-pass to do the post-pass. Finally, note that although the symmetrization (setting $S$ and $T$ sharding equal) assumption is discussed at the end of Section 7.1, we do not assume this property for the scheme presentation.

**Pre-pass**    At layer $l$, each CompNode$_i$ starts with the $R_i-$sharded hidden states $\boldsymbol{h}_i^R$. If necessary for the attention block, it performs layer normalization on these states. Then, it performs $Q, K, V$-projection to

get the $R_i$−sharded query, key and value states $\boldsymbol{q}_i^R, \boldsymbol{k}_i^R, \boldsymbol{v}_i^R$. CompNode$_i$ then applies rotary or positional embedding to $\boldsymbol{q}_i^R, \boldsymbol{k}_i^R$, using sharded positional embeddings $\boldsymbol{p}_i$ (the node can generate these upon setup to avoid any communication overhead, since it only depends on its index set $R_i$), and returns all of $\boldsymbol{q}_i^R, \boldsymbol{k}_i^R, \boldsymbol{v}_i^R$, as described in Algorithm 3.

**Attention-pass** After the pre-pass, each AttnNode$_{jk}$ receives shards $\boldsymbol{q}_{ij}^{RS}, \boldsymbol{k}_{ik}^{RT}, \boldsymbol{v}_{ik}^{RT}$ from each CompNode$_i$, which are slices of its pre-pass outputs $\boldsymbol{q}_i^R, \boldsymbol{k}_i^R, \boldsymbol{v}_i^R$ along the token dimension. By concatenating the rows of $\boldsymbol{q}_{ij}^{RS}, \boldsymbol{k}_{ik}^{RT}, \boldsymbol{v}_{ik}^{RT}$ over all $1 \leq i \leq \alpha$ in a particular order, AttnNode$_{jk}$ obtains $\boldsymbol{q}_j^S, \boldsymbol{k}_k^S, \boldsymbol{v}_k^S$. Precisely, the order for $\boldsymbol{q}_j^S$ is determined by the order in which one should concatenate elements of sorted sets $R_i \cap S_j$ over all $1 \leq i \leq \alpha$, to obtain sorted $S_j$; and likewise for $\boldsymbol{k}_k^S, \boldsymbol{v}_k^S$, it is the order in which one concatenates elements of sorted sets $R_i \cap T_k$ over all $1 \leq i \leq \alpha$, to get sorted $T_k$. Next, in grouped-query attention, the key and value heads should be repeated to match query heads. Then AttnNode$_{jk}$ can compute $\boldsymbol{q}_j^S(\boldsymbol{k}_k^T)^T + \boldsymbol{m}_{jk}^{ST}$, where matrix multiplication is performed per-head and the attention mask is broadcasted. This results in the submatrix $\boldsymbol{a}_{jk}^{ST} = \boldsymbol{a}[:, S_j, T_k]$ of the masked attention logits $\boldsymbol{a}$. For use in the post-pass, AttnNode$_{jk}$ also stores sums of exponentials[4] of elements of the submatrix rows, which was defined in Section 7.1 as $\boldsymbol{e}_{jk}^{ST}$. Finally, AttnNode$_{jk}$ takes the row-wise softmax and performs value multiplication to get $\boldsymbol{u}_{jk}^{ST} = \text{softmax}(\boldsymbol{a}_{jk}^{ST})\boldsymbol{v}_k^T$. Both $\boldsymbol{u}_{jk}^{ST}, \boldsymbol{e}_{jk}^{ST}$ are returned, as in Algorithm 4.

**Post-pass** Finally, after the attention-pass, each CompNode$_i$ receives $\boldsymbol{u}_{ijk}^{RST}, \boldsymbol{e}_{ijk}^{RST}$ from each AttnNode$_{jk}$, which are slices of its attention-pass outputs $\boldsymbol{u}_{jk}^{ST}, \boldsymbol{e}_{jk}^{ST}$ along the token dimensions. Then, for each fixed $1 \leq k \leq \beta$, CompNode$_i$ concatenates the rows of $\boldsymbol{u}_{ijk}^{RST}, \boldsymbol{e}_{ijk}^{RST}$ over all $1 \leq j \leq \beta$ to obtain $\boldsymbol{u}_{ik}^{RT}, \boldsymbol{e}_{ik}^{RT}$, in a similar manner as the attention-pass concatenation. Now, the order is determined by the order of concatenation of elements of sorted $R_i \cap S_j \cap T_k$ over all $1 \leq j \leq \beta$ to form sorted $R_i \cap T_k$. Next, CompNode$_i$ aims to combine these results $\boldsymbol{u}_{ik}^{RT}, \boldsymbol{e}_{ik}^{RT}$ over $1 \leq k \leq \gamma$, into the $R_i$−sharded (pre $O$-proj) output of attention, which equals

$$\text{softmax}(\boldsymbol{a})[:, R_i]\boldsymbol{v} = \sum_{k=1}^{\gamma} \text{softmax}(\boldsymbol{a})[:, R_i, T_k]\boldsymbol{v}[:, T_k].$$

by blocked matrix multiplication. The terms in the summation are not known to CompNode$_i$, since indexing here is performed post-softmax. To correct for this, observe that for a row vector $v \in \mathbb{R}^N$ and any $1 \leq k \leq \gamma$,

$$\text{softmax}(v)[S_k] = \frac{\text{expsum}(v[S_k])}{\sum_{k'} \text{expsum}(v[S_{k'}])} \cdot \text{softmax}(v[S_k]).$$

Thus, the above summation can be simplified to

$$\text{softmax}(\boldsymbol{a})[:, R_i]\boldsymbol{v} = \frac{\sum_k \text{expsum}(\boldsymbol{a}[:, R_i, T_k]) \odot (\text{softmax}(\boldsymbol{a}[:, R_i, T_k])\boldsymbol{v}[:, T_k])}{\sum_k \text{expsum}(\boldsymbol{a}[:, R_i, T_k])}$$

$$= \frac{\sum_k \text{expsum}(\boldsymbol{a}_{ik}^{RT}) \odot (\text{softmax}(\boldsymbol{a}_{ik}^{RT})\boldsymbol{v}_k^T)}{\sum_k \text{expsum}(\boldsymbol{a}_{ik}^{RT})}$$

$$= \frac{\sum_k \boldsymbol{e}_{ik}^{RT} \odot \boldsymbol{u}_{ik}^{RT}}{\sum_k \boldsymbol{e}_{ik}^{RT}}$$

by using the defined notation. Here, the fraction is elementwise division, and expsum is performed row-wise along the last dimension. Now, each aggregate term in the numerator and denominator summations is known to the CompNode. In essence, the CompNode is performing a weighted average of concatenated AttnNode $\boldsymbol{u}$ results, with the weights also coming from AttnNodes $\boldsymbol{e}$ results. To get the final output of attention corresponding to row (token) indices in $R_i$, it finally performs $O$-projection. See Algorithm 5 for an implementation of this procedure.

---

[4]Due to precision and overflow issues with the expsum, we actually store the maximum of each submatrix row, subtract it from the row, and then compute the expsum. Then both the row-wise expsum and maximum are returned, instead of just $\boldsymbol{e}_{jk}^{ST}$.

---

**Algorithm 2** Cascade Single Layer Forward Pass

---

**input** $R_i-$sharded layer $l$ hidden states $\boldsymbol{h}_i^R \in \mathbb{R}^{|R_i| \times d_{emb}}$ in CompNode$_i$, for each $1 \leq i \leq \alpha$
**output** $R_i-$sharded layer $l+1$ hidden states $\boldsymbol{h}_i^R \in \mathbb{R}^{|R_i| \times d_{emb}}$ in CompNode$_i$, for each $1 \leq i \leq \alpha$
 1: **for** $i = 1$ to $\alpha$: CompNode$_i$ **do**
 2:    $\boldsymbol{q}_i^R, \boldsymbol{k}_i^R, \boldsymbol{v}_i^R \leftarrow$ **pre_pass**$(\boldsymbol{h}_i^R)$ {Algorithm 3}
 3:    **for** $j, k = 1$ to $\beta$ **do**
 4:       $\boldsymbol{q}_{ij}^{RS}, \boldsymbol{k}_{ik}^{RT}, \boldsymbol{v}_{ik}^{RT} \leftarrow$ comp_qkv_slice$(\boldsymbol{q}_i^R, \boldsymbol{k}_i^R, \boldsymbol{v}_i^R)$ {$S, T-$slicing along row token dimension}
 5:       Send $\boldsymbol{q}_{ij}^{RS}, \boldsymbol{k}_{ik}^{RT}, \boldsymbol{v}_{ik}^{RT}$ to AttnNode$_{jk}$
 6:    **end for**
 7: **end for**
 8: **for** $j, k = 1$ to $\beta$: AttnNode$_{jk}$ **do**
 9:    $\boldsymbol{q}_j^S, \boldsymbol{k}_k^T, \boldsymbol{v}_k^T \leftarrow$ attn_qkv_concat$\{\boldsymbol{q}_{ij}^{RS}, \boldsymbol{k}_{ik}^{RS}, \boldsymbol{v}_{ik}^{RT}\}_{i=1}^{\alpha}$ {$R-$interleaved concatenation of rows}
10:    $\boldsymbol{u}_{jk}^{ST}, \boldsymbol{e}_{jk}^{ST} \leftarrow$ **attn_pass**$(\boldsymbol{q}_j^S, \boldsymbol{k}_k^T, \boldsymbol{k}_k^T)$ {Algorithm 4}
11:    **for** $i = 1$ to $\alpha$ **do**
12:       $\boldsymbol{u}_{ijk}^{RST}, \boldsymbol{e}_{ijk}^{RST} \leftarrow$ attn_ue_slice$(\boldsymbol{u}_{jk}^{ST}, \boldsymbol{e}_{jk}^{ST})$ {$R-$slicing along row token dimension}
13:       Send $\boldsymbol{u}_{ijk}^{RST}, \boldsymbol{e}_{ijk}^{RST}$ to CompNode$_i$
14:    **end for**
15: **end for**
16: **for** $i = 1$ to $\alpha$: CompNode$_i$ **do**
17:    **for** $k = 1$ to $\beta$ **do**
18:       $\boldsymbol{u}_{ik}^{RT}, \boldsymbol{e}_{ik}^{RT} \leftarrow$ comp_ue_concat$\{\boldsymbol{u}_{ijk}^{RST}, \boldsymbol{e}_{ijk}^{RST}\}_{j=1}^{\beta}$ {$S-$interleaved concatenation of rows}
19:    **end for**
20:    $\boldsymbol{o}_i^R \leftarrow$ **post_pass**$\{\boldsymbol{u}_{ik}^{RT}, \boldsymbol{e}_{ik}^{RT}\}_{k=1}^{\beta}$ {Algorithm 5}
21:    $\boldsymbol{h}_i^R \leftarrow \boldsymbol{h}_i^R + \boldsymbol{o}_i^R$ {Residual connection}
22:    $\boldsymbol{h}_i^R \leftarrow$ **mlp**$(\boldsymbol{h}_i^R)$ {Standard MLP block pass}
23: **end for**

---

**Algorithm 3** CompNode$_i$ Single Layer Pre-Pass

---

**input** $R_i-$sharded hidden states $\boldsymbol{h}_i^R \in \mathbb{R}^{|R_i| \times d_{emb}}$
    $R_i-$sharded position embeds $\boldsymbol{p}_i^R \in \mathbb{R}^{|R_i| \times d}$
**output** $R_i-$sharded query states $\boldsymbol{q}_i^R \in \mathbb{R}^{H \times |R_i| \times d}$
    $R_i-$sharded key/value states $\boldsymbol{k}_i^R, \boldsymbol{v}_i^R \in \mathbb{R}^{H_{KV} \times |R_i| \times d}$
 1: $\boldsymbol{q}_i^R \leftarrow$ q_proj$(\boldsymbol{h}_i^R)$
 2: $\boldsymbol{k}_i^R \leftarrow$ k_proj$(\boldsymbol{h}_i^R)$
 3: $\boldsymbol{v}_i^R \leftarrow$ v_proj$(\boldsymbol{h}_i^R)$
 4: $\boldsymbol{q}_i^R \leftarrow$ rotary_pos_emb$(\boldsymbol{q}_i^R, \boldsymbol{p}_i^R)$
 5: $\boldsymbol{k}_i^R \leftarrow$ rotary_pos_emb$(\boldsymbol{k}_i^R, \boldsymbol{p}_i^R)$
 6: **return** $\boldsymbol{q}_i^R, \boldsymbol{k}_i^R, \boldsymbol{v}_i^R$

---

---

**Algorithm 4** $\text{AttnNode}_{jk}$ Single Layer Attention-Pass

---

**input** $S_j-$sharded query states $\boldsymbol{q}_j^S \in \mathbb{R}^{H \times |S_j| \times d}$

$T_k-$sharded key/value states $\boldsymbol{k}_k^T, \boldsymbol{v}_k^T \in \mathbb{R}^{H_{KV} \times |T_k| \times d}$

$(S_j, T_k)-$sharded attn mask $\boldsymbol{m}_{jk}^{ST} \in \mathbb{R}^{H \times |S_j| \times |T_k|}$

**output** $(S_j, T_k)-$sharded attn vals $\boldsymbol{u}_{jk}^{ST} \in \mathbb{R}^{H \times |S_j| \times d}$

$(S_j, T_k)-$sharded attn score expsums $\boldsymbol{e}_{jk}^{ST} \in \mathbb{R}^{H \times |S_j|}$

1: $\boldsymbol{k}_k^T \leftarrow \text{repeat\_kv}(\boldsymbol{k}_k^T)$
2: $\boldsymbol{v}_k^T \leftarrow \text{repeat\_kv}(\boldsymbol{v}_k^T)$
3: $\boldsymbol{a}_{jk}^{ST} \leftarrow \boldsymbol{q}_j^S (\boldsymbol{k}_k^T)^T + \boldsymbol{m}_{jk}^{ST}$
4: $\boldsymbol{a}_{jk}^{ST} \leftarrow \exp(\boldsymbol{a}_{jk}^{ST})$
5: $\boldsymbol{e}_{jk}^{ST} \leftarrow \text{sum}(\boldsymbol{a}_{jk}^{ST})$
6: $\boldsymbol{a}_{jk}^{ST} \leftarrow \boldsymbol{a}_{jk}^{ST} / \boldsymbol{e}_{jk}^{ST}$
7: $\boldsymbol{u}_{jk}^{ST} \leftarrow \boldsymbol{a}_{jk}^{ST} \boldsymbol{v}_k^T$
8: **return** $\boldsymbol{u}_{jk}^{ST}, \boldsymbol{e}_{jk}^{ST}$

---

**Algorithm 5** $\text{CompNode}_i$ Single Layer Post-Pass

---

**input** $(R_i, T_k)-$sharded attn vals $\boldsymbol{u}_{ik}^{RT} \in \mathbb{R}^{H \times |R_i| \times d}$ for all $1 \leq k \leq \gamma$

$(R_i, T_k)-$sharded attn score expsums $\boldsymbol{e}_{ik}^{RT} \in \mathbb{R}^{H \times |R_i|}$ for all $1 \leq k \leq \gamma$

**output** $R_i-$sharded attn output $\boldsymbol{o}_i^R \in \mathbb{R}^{|R_i| \times d_{emb}}$

1: Initialize $\boldsymbol{e}_i^R \in \mathbb{R}^{H \times |R_i|}$ with zeroes
2: **for** $k = 1$ to $\gamma$ **do**
3: $\quad \boldsymbol{e}_i^R \leftarrow \boldsymbol{e}_i^R + \boldsymbol{e}_{ik}^{RT}$
4: **end for**
5: Initialize $\boldsymbol{o}_i^R \in \mathbb{R}^{|R_i| \times d_{emb}}$ with zeroes
6: **for** $k = 1$ to $\gamma$ **do**
7: $\quad \boldsymbol{w}_{ik}^{RT} \leftarrow \boldsymbol{e}_{ik}^{RT} / \boldsymbol{e}_i^R$
8: $\quad \boldsymbol{o}_i^R \leftarrow \boldsymbol{o}_i^R + \boldsymbol{w}_{ik}^{RT} \odot \boldsymbol{u}_{ik}^{RT}$
9: **end for**
10: $\boldsymbol{o}_i^R \leftarrow \text{o\_proj}(\boldsymbol{o}_i^R)$
11: **return** $\boldsymbol{o}_i^R$

---

## G    REVERSAL ANALYSIS ON LAYERS & LLAMA

In Section 7.2.1, we described experiments performed on the hidden states of Gemma-2-2B-IT, where a bidirectional-attention model was trained to reverse the sharded hidden states into the original text prompt. In Table 3, we showed that the hiddens are largely secure to this attack for a suitable choice of $c$ and $\alpha$.

In this section, we first analyze if this is also true for Llama 3.1 8B-Instruct. We run a similar experimental setup as described in Section 7.2.1, except we use Llama hidden representations, and we also use it as the reversal model; this also therefore tests if increasing the reversal model's capacity is a suitable method for improving sharded reconstruction. Due to the computational constraints of training with a larger model, we examine this only for $c = 8, \alpha = 8$ and $c = 8, \alpha = 12$. The reconstruction ROUGE scores are 0.1718 and 0.1443 respectively, significantly lower than those obtained with the same parameters for Gemma. We leave to future work the interesting question of whether this implies that Llama representations are inherently more resistant to decoding than Gemma representations.

Next, we analyze whether our results hold irrespective of the layer of the model used. We run additional experiments on the hiddens of layers 11 and 21 of Gemma-2-2B-IT. Our results are shown in Table 11. We see that there is no substantial difference in ROUGE score as the layer changes.

Table 11: ROUGE scores of text reconstruction from the hiddens of various layers of Gemma-2-2B-IT for different $(c, \delta)$-sharding setups. We see that the reconstruction quality is similar across layers.

| **Layer** | $c = \alpha = 4$ | $c = \alpha = 8$ |
|:---:|:---:|:---:|
| 1 | 0.4268 | 0.2218 |
| 11 | 0.4627 | 0.2467 |
| 21 | 0.4021 | 0.2158 |

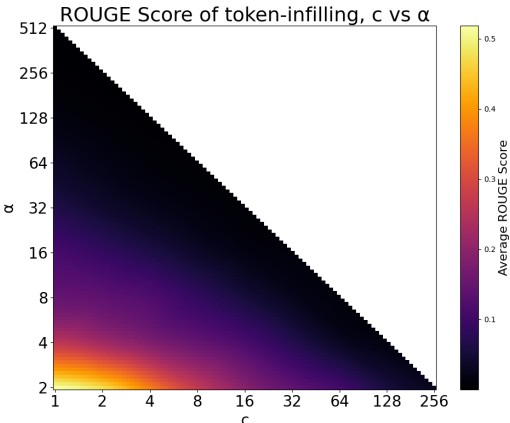

Figure 1: ROUGE scores for Layer $0$ token prediction using ModernBERT-Large, as a function of $c$, the number of 'clusters' in the sharding scheme, and $\alpha$, the number of CompNodes. Higher $\alpha$ and higher $c$ tend towards lower ROUGE, increasing security.

## H  LAYER 0 SECURITY CONTINUED

Here we continue the discussion on the security of CompNodes at Layer 0, as begun in Section 7.2.1.

Given that a CompNode has access to some of the $N$ token embeddings, say $[e_{n_1}, e_{n_2}, \ldots, e_{n_t}]$, the possibility of reconstruction of the full prompt $x$ is theoretically lower bounded by the entropy of the distribution

$$p(\{x_j : j \in [N] \setminus \{n_1, \ldots, n_t\}\} \mid x_{n_1}, x_{n_2}, \ldots, x_{n_t})$$

The true distribution cannot easily be computed, but we may approximate it using masked token infilling, as utilized in training models like BERT (Devlin et al., 2019), RoBERTa (Liu et al., 2019), and ModernBERT (Warner et al., 2024). We use the recently released state-of-the-art ModernBERT-large to probe properties of this distribution under a $(c, \delta)$-sharding scheme. We use 200 samples from FineWeb-Edu to compute the ROUGE (Lin, 2004) score over argmax-token generation. Our results are shown in Figure 1. We see that ROUGE score diminishes as both $c$ and $\alpha$ increase; good security seems to be achieved for $c, \alpha \geq 8$ (correspondingly, $\delta \geq 64$).

# I $S$-SHARDING SECURITY EXPERIMENTS

In Section 7.2.1, we examined the security of the CompNodes in the case where their $R$ sharding is precisely $(c, \delta)$-sharding. We now turn our attention to the security of AttnNodes from $S$ and $T$ sharding, to justify our assumption from Section 7.1 that these can be made symmetric. Recall that $\{S_j\}_{j=1}^{\beta}$ and $\{T_k\}_{k=1}^{\gamma}$ are shardings of $[N]$ that must satisfy that all index pairs in $[N] \times [N]$ are covered by some $S_j \times T_k$. Observe that a node holding the $j$th and $k$th query and $j$th key rows has – in the worst case – the same information as if they hold the $j$th query and $k$th key rows. This is because the query and key matrices are linear projections of the same hidden states; and at Layer 0, an adversary can exactly reverse them to these hidden states, by passing the entire vocabulary through these linear projections and matching, akin to the vocab-matching attack. Even in later layers, if the linear projection is reversible (injective), this worst case occurs. Therefore, we justify our simplification in Section 7.1 that $T$ sharding may be set equal to $S$ sharding without security loss.

One simple way to satisfy the pairwise-coverage requirement of $S$ sharding, given a particular choice of $R$ sharding, is to set $\beta = \alpha$ and $S$ sharding equal to $R$ sharding, and have each AttnNode$_{jk}$ receive the union of query shards from CompNode$_j$ and key/value shards from CompNode$_k$. However, this results in each AttnNode having twice as many indices as each CompNode. To improve security, we instead propose a further $m-$split as follows. We set $\beta = m\alpha$, and further partition each $R_i$ into $m$ shards $R_{i,1}, \ldots, R_{i,m}$. Then we let $S_{m(i-1)+l} = R_{i,l}$ for all $1 \leq i \leq \alpha$ and $1 \leq l \leq m$. This ensures that pairwise coverage is maintained, but reduces the number of indices that each AttnNode has access to. For general $m$, the ratio of AttnNode to CompNode indices is given by $\frac{2}{m}$, assuming each split of $R_i$ into $m$ shards is uniform. Using this split increases the value of $\beta^2$, the total number of AttnNodes, by a factor of $m^2$.

There still remains a degree of freedom in deciding the exact choice of subdividing $R_i$ into the subsets $R_{i,k}$. Under the assumption that $R_i$ follows a $(c, \delta)$-sharding, we propose that $R_{i,l}$ contains the elements of (sorted) $R_i$ at indices $l, l + \delta, \ldots, l + (t - 1)\delta$, where $t = \frac{N}{c\alpha}$ and the split factor is $m = c$. This scheme for $R_i$ takes the elements of the $(c, \delta)$-sequence and spreads them out between the associated $R_{i,l}$.

For example, consider the case where $c = 3, \delta = 9, N = 18$. Then $\alpha = 3$, and the split factor is $m = 3$, so we get:

$$R_1 = \{1, 2, 3, 10, 11, 12\} \quad R_2 = \{4, 5, 6, 13, 14, 15\} \quad R_3 = \{7, 8, 9, 16, 17, 18\}$$
$$R_{1,1} = \{1, 10\} \quad\quad\quad R_{1,2} = \{2, 11\} \quad\quad\quad R_{1,3} = \{3, 12\}$$
$$R_{2,1} = \{4, 13\} \quad\quad\quad R_{2,2} = \{5, 14\} \quad\quad\quad R_{2,3} = \{6, 15\}$$
$$R_{3,1} = \{7, 16\} \quad\quad\quad R_{3,2} = \{8, 17\} \quad\quad\quad R_{3,3} = \{9, 18\}$$
$$S_{1121} = \{1, 4, 10, 13\} \quad S_{1122} = \{1, 5, 10, 14\} \quad S_{1123} = \{1, 6, 10, 15\}$$
$$S_{1221} = \{2, 4, 11, 13\} \quad S_{1222} = \{2, 5, 11, 14\} \quad S_{1223} = \{2, 6, 11, 15\}$$
$$S_{1321} = \{3, 4, 12, 13\} \quad S_{1322} = \{3, 5, 12, 14\} \quad S_{1323} = \{3, 6, 12, 15\}$$
$$S_{2131} = \{4, 7, 13, 16\} \quad S_{2132} = \{4, 8, 13, 17\} \quad S_{2133} = \{4, 9, 13, 18\}$$
$$S_{2231} = \{5, 7, 14, 16\} \quad S_{2232} = \{5, 8, 14, 17\} \quad S_{2233} = \{5, 9, 14, 18\}$$
$$S_{2331} = \{6, 7, 15, 16\} \quad S_{2332} = \{6, 8, 15, 17\} \quad S_{2333} = \{6, 9, 15, 18\}$$
$$S_{1131} = \{1, 7, 10, 16\} \quad S_{1132} = \{1, 8, 10, 17\} \quad S_{1133} = \{1, 9, 10, 18\}$$
$$S_{1231} = \{2, 7, 11, 16\} \quad S_{1232} = \{2, 8, 11, 17\} \quad S_{1233} = \{2, 9, 11, 18\}$$
$$S_{1331} = \{3, 7, 12, 16\} \quad S_{1332} = \{3, 8, 12, 17\} \quad S_{1333} = \{3, 9, 12, 18\}$$

Here, we use the shorthand notation $S_{ili'l'} = R_{i,l} \cup R_{i',l'} = S_{m(i-1)+l} \cup S_{m(i'-1)+l'}$, which is precisely the set of indices that AttnNode$_{j,k}$ has access to for $j = m(i - 1) + l, k = m(i' - 1) + l'$. Sharding $S$ in this way retains the security desiderata we care about with respect to the vocab-matching attack. Since each $R_{i,l}$ has elements that are separated by $\delta$, and each $S_{ili'l'}$ combines elements from two different $R_{i,l}$, then we can never have more than 2 consecutive elements in $S_{ili'l'}$. Furthermore, the largest number of missing tokens between two elements of $S_{ili'l'}$ (i.e. the largest 'gap') is lower bounded by $\frac{\delta}{2}$. Therefore, for sufficiently large $\delta$, the sharding is secure to the vocab-matching attack.

To test security against learning-based reversal attacks, we conducted experiments with the above scheme for $m = \{2, 3, 4\}$, with $c = 8$ and $\alpha = 8$ (and so $\delta = 64$). Due to computational constraints, we focus our experiments on Gemma-2-2B-IT on layer 1; we expect similar trends for Llama-3.1-8B-Instruct and for other layers. We train a single model for all shard possibilities that arise from $S_{ili'l'}$. Experiments are conducted with the same dataset and model setup as described in Section 7.2.1. Our results are shown in Table 12. We

see that although $m = 2$ results in a relatively higher ROUGE than that for the CompNodes in Table 3 (0.3057 vs 0.222), the score for $m = 4$ is very similar; therefore, we recommend using at least $m = 4$ for security.

Table 12: ROUGE scores for different values of splitting parameter $m$ on layer 1 of Gemma-2-2B-IT with $c = 8, \alpha = 8$. We see that the score for $m = 4$ is similar to that of CompNodes in Table 3 for the same $c$ and $\alpha$.

| $m$ | ROUGE |
|-----|--------|
| 2 | 0.3057 |
| 3 | 0.2643 |
| 4 | 0.2376 |

## J COMPUTATIONAL OVERHEAD ANALYSIS

In Section 7.3.1, we claimed that Cascade has little overhead in computational costs compared to vanilla inference. We justify this statement in the analysis below, by comparing CompNode and AttnNode steps against the vanilla forward pass. For simplicity of analysis, we assume symmetry of $S$ and $T$ sharding, as justified in Appendix I.

Indeed, most operations performed by CompNodes will treat the (row) token dimension as the batch dimension. In the pre-pass, these are normalization and $Q, K, V$-projection; and in the post-pass, these are attention value compilation, $O$-projection, residual connection, and the MLP block. Except for attention value compilation, these steps all occur in the vanilla pass, so the CompNodes combined will perform the exact same operations as in vanilla inference: there are no extra computations performed.

The only extra operations thus come from **(a)** attention value compilation (linear weighting of partial attention outputs) by CompNodes in the post-pass, and **(b)** AttnNode floating point computations which do not appear in the vanilla pass, i.e. expsums of shards of attention score rows. This is because all other steps of the Cascade self-attention either treat the tokens as batch elements, or involve splitting up matrix multiplication into multiplication of sharded matrices; and the latter is blocked matrix multiplication, which does not inherently change the operations performed.

Now, **(a)** only involves $\sim H|R_i|d$ operations for each CompNode$_i$, since it involves a few steps of elementwise summation and multiplication of $H \times |R_i| \times d$ matrices (after broadcasting). Summing this over all $1 \leq i \leq \alpha$ gives $\sim \sum_i H|R_i|d = HNd$ extra operations. Also, **(b)** only involves $\sim H|S_j||S_k|$ operations for each AttnNode$_{jk}$ because expsum is done over rows of an $H \times |S_j| \times |S_k|$ shard of attention scores. Summing over all $1 \leq j, k \leq \beta$, we see this requires $\sim \sum_{j,k} H|S_j||S_k| = HN^2$ extra operations in total. This means the total AttnNode computation overhead is $\sim HN(d + N)$ operations.

Importantly, this is much cheaper than most computation-heavy steps in standard inference. Compared to the $\sim HN^2d$ operations from multiplication of $H \times N \times N$ attention scores with $H \times N \times d$ values, this overhead requires $\sim \frac{1}{N} + \frac{1}{d}$ times as many operations. Since $d$ is often in the hundreds, we can ensure for large $N$, say $N \geq 256$, that this ratio is quite small. Furthermore, if $N$ is not large, then the overhead is still negligible compared to the $\sim HNd_{emb}d$ operations from $Q, K, V$-projection, since it requires $\sim \frac{1}{d_{emb}} + \frac{N}{d_{emb}}$ times as many operations and $d_{emb}$ is in the hundreds or thousands. Essentially, the choice of sharding does not significantly affect the total computational overhead, and this overhead is negligible compared to computation performed in a vanilla forward pass.

## K COMMUNICATION ANALYSIS

In Section 7.3.1, we gave the total communication byte and time overheads for performing a inference on a single layer of an LLM with Cascade. Here, we provide a full justification of these equations. Like in Appendix J, we assume symmetry of $S$ and $T$ sharding, so the superscript $T$ in sharded notation is replaced with $S$.

Recall that in each layer, there are two communication stages: **(A)** the CompNodes send sharded query, key, value matrices to the AttnNodes between pre-pass and attention-pass, and **(B)** the AttnNodes send sharded attention outputs and expsums to the CompNodes between attention-pass and post-pass. We operate under the

assumption that all CompNodes synchronize before **(A)** and all AttnNodes synchronize before **(B)**, so that we can derive an exact expression for communication cost; this makes our communication cost derivation a worst-case analysis. See Appendix N for optimizations that can be made if this assumption is relaxed.

For single-layer inference, in stage **(A)**, CompNode$_i$ must send each of the $|R_i|$ rows of the $H \times |R_i| \times d$ query matrix $\boldsymbol{q}_i^R$ to some AttnNodes. In particular, for a row index $r \in R_i$, it sends the row $\boldsymbol{q}[:, r, :]$ of $\boldsymbol{q}_i^R$ to all AttnNodes$_{j_r, k}$ with $1 \leq k \leq \beta$, where $j_r$ is the unique index satisfying $r \in S_{j_r}$. That is, CompNode$_i$ sends each of its $|R_i|$ rows to exactly $\beta$ AttnNodes. Since each row contains $Hd$ elements, then CompNode$_i$ must send out $\beta|R_i|Hd$ elements from sharded query states. A similar analysis shows CompNode$_i$ sends out $2\beta|R_i|H_{KV}d$ elements from sharded key and value states, so it sends out a total of $\beta|R_i|d(H + 2H_{KV})$ elements. Summing this over all $i$ and noting $\sum_{i=1}^{\alpha} |R_i| = N$ gives the total bytes communicated in **(A)**:

$$\text{CommBytes}_A = \beta F d (H + 2H_{KV}) \cdot N.$$

Assuming perfect parallel transport and uniform bandwidth $B$ across nodes, i.e. all communication overhead comes from CompNode with the most elements to send (plus latency $\tau$), the communication time in stage **(A)** is

$$\text{CommTime}_A = \tau + \frac{\beta F d (H + 2H_{KV})}{B} \cdot \max_i |R_i|.$$

Next, in stage **(B)**, each AttnNode$_{jk}$ must send each CompNode$_i$ some rows of its partial post-value attention outputs $\boldsymbol{u}_{jk}^{SS}$ and attention score row expsums $\boldsymbol{e}_{jk}^{SS}$. These matrices[5] are of shapes $H \times |S_j| \times d$ and $H \times |S_j| \times 2$, respectively, and CompNode$_i$ receives $|R_i \cap S_j|$ out of the $|S_j|$ rows from each. This means the total number of elements that AttnNode$_{jk}$ sends to all CompNodes is

$$(d + 2)H \cdot \sum_{i=1}^{\alpha} |R_i \cap S_j| = (d + 2)H \cdot |S_j|.$$

Since $\sum_{j,k=1}^{\beta} |S_j| = \beta \sum_{j=1}^{\beta} |S_j| = \beta N$, this means the total number of bytes sent by all $\beta^2$ AttnNodes is

$$\text{CommBytes}_B = \beta F (d + 2) H \cdot N.$$

And, again under the parallel transport and uniform bandwidth assumption, the communication time in **(B)** is

$$\text{CommTime}_B = \tau + \frac{F(d + 2)H}{B} \cdot \max_j |S_j|.$$

Combining these costs, we obtain the following total communication byte and time overheads for a single layer:

$$\text{CommBytes} = \beta F (2dH + 2dH_{KV} + 2H) \cdot N,$$
$$\text{CommTime} = 2\tau + \frac{\beta F d (H + 2H_{KV})}{B} \cdot \max_i |R_i| + \frac{F(d + 2)H}{B} \cdot \max_j |S_j|.$$

Finally, we compute the number of communication rounds per layer. Stage **(A)** has each of the $\alpha$ CompNodes send results to at most $\beta^2$ AttnNodes, which is at most $\alpha\beta^2$ rounds. Stage **(B)** has each of the $\beta^2$ AttnNodes send results to at most $\alpha$ CompNodes, which is at most $\alpha\beta^2$ rounds. In total, the rounds per layer are bounded above by $2\alpha\beta^2$. This can be quite large, but we can guarantee a tighter upper bound if our scheme involves $(c, \delta)$-sharding for CompNodes with $m$-splitting of AttnNodes (as in Appendix I). Here, $\beta = m\alpha$ since each of the $\alpha$ shards in $\{R_i\}_{i=1}^{\alpha}$ is split into $m$ pieces to form $\{S_j\}_{j=1}^{\beta}$. Each CompNode sends results to $m\beta$ AttnNodes, and each AttnNode sends results to 1 CompNode, so there are $m\alpha\beta + \beta^2 = 2\beta^2$ rounds. Essentially, the number of rounds scales linearly with the number of AttnNodes.

## L   METHODOLOGY OF TOTAL RUNTIME & COMMUNCIATION BYTES BENCHMARK

In Section 7.3.2, we presented our results comparing Cascade against two recent SMPC schemes, MPCFormer Li et al. (2023a) and Puma Dong et al. (2023a), on Bert-Base and Bert-Large for *total runtime* and *total communicated bytes*. Here, we describe further the methodology used to populate Table 4.

---

[5]See the footnote in the post-pass section of Appendix F. In practice, we store a maximum and shifted expsum (i.e. subtract the maximum from the row, then take expsum) per row instead of just an expsum, for numerical stability in later computations. This is why the last dimension of $\boldsymbol{e}_{jk}^{SS}$ is 2 instead of 1.

The total communicated bytes can be computed by plugging in model parameters $F, d, H, H_{KV}$, networking parameters $B, \tau$, and sharding-related quantities $\max_i |R_i|, \max_j |S_j|, \beta$ into the single-layer communicated byte formula (Equation (1)), and multiplying by the number of layers $L$. The model parameters are as follows: for Bert-Base, we have $L = 12$ layers, $d = 64$, and $H = H_{KV} = 12$, and in Bert-Large, we have $L = 24$ layers, $d = 64$, and $H = H_{KV} = 16$. In both models, the default float32 quantization is $F = 4$ bytes. The networking and sharding parameters are described at the end of this section.

To compute our total runtime, we similarly calculate the single-layer communication time by Equation (2), multiply it by the number of layers $L$, and add a worst-case $2\tau$ latency for the user sending the initial message and receiving logits. For inference time, we run 368 LiveBench (White et al., 2024) math prompts on Bert-Base and Bert-Large, and take the maximum of these runtimes as a worst-case estimate. Total runtime is then calculated by summing these two values. For fair comparison against costs from Dong et al. (2023a), which appear in the first two rows of Table 4, we match against the Puma experiment settings. We truncate or pad all prompts to $N = 128$ tokens. On the hardware side, our experiments are run on a Paperspace machine with 8 vCPU and 44GB RAM, with the CPU model being Intel Xeon Gold 5315Y @ 3.20GHz. We use Ubuntu 22.04.2 LTS with Kernel 5.15.0-125-generic. This is a fair setup for comparison to Puma's numbers: they use a CPU setup with Intel Xeon (Ice Lake) Platinum 8369B @ 2.70GHz, for 32 vCPU and 128 GB RAM. They similarly use Ubuntu 20.04.6 LTS with Linux kernel 5.4.0-144-generic.

For networking parameters, we simulate a uniform inter-node bandwidth $B = 5$Gbps and round-trip latency $\tau = 1$ms, like Puma. We use two Cascade setups with $(c, \delta)$-sharding and $m$-splitting of AttnNodes: $\alpha = 4$, $c = 8$, $\beta = 16$, $m = 4$, $\delta = 32$, and $\alpha = 8$, $c = 8$, $\beta = 32$, $m = 4$, $\delta = 64$. The latter was chosen as its security is strongly supported by additional experiments in Appendix I, and the former was chosen as a similar sharding scheme with fewer nodes.

## M   FULL INFORMATION LEAKAGE ANALYSIS

To rigorously analyze information leakage, we examine *isolated* computational stages of nodes. Like in Appendix J and Appendix K, for simplicity of analysis, we assume $S$ and $T$ sharding are equal.

In Cascade, whenever a node (CompNode or AttnNode) begins a stage of computation (e.g. pre-pass, attention-pass, or post-pass) where it receives no information *during* the computation, all new input leakage at that stage comes from the information they receive at stage initialization. This means all leakage comes from the following shards, at each layer:

$$\text{CompNode}_i \rightarrow \boldsymbol{h}_i^R, \boldsymbol{u}_{i1}^{RS}, \boldsymbol{e}_{i1}^{RS}, \ldots, \boldsymbol{u}_{i\beta}^{RS}, \boldsymbol{e}_{i\beta}^{RS}$$
$$\text{AttnNode}_{jk} \rightarrow \boldsymbol{q}_j^S, \boldsymbol{k}_k^S, \boldsymbol{v}_k^S.$$

We have already examined information leakage from $\boldsymbol{h}_i^R$ in great detail in Section 7.2.1. We also discussed information leakage from $\boldsymbol{q}_j^S, \boldsymbol{k}_k^S, \boldsymbol{v}_k^S$ in Appendix I: in the worst-case, this reveals the same information as hidden states $\boldsymbol{h}_j^S, \boldsymbol{h}_k^S$, i.e. the AttnNode has hidden states at indices $S_j \cup S_k$ and thus twice as much information as CompNodes; and we discussed how $m$-splitting of AttnNodes could alleviate this security concern, by further restricting AttnNode information.

So, for a comprehensive security analysis, all that remains is to consider leakage from $\boldsymbol{u}_{i1}^{RS}, \boldsymbol{e}_{i1}^{RS}, \ldots, \boldsymbol{u}_{i\beta}^{RS}, \boldsymbol{e}_{i\beta}^{RS}$, the information that CompNode$_i$ receives from $\beta$ AttnNodes to begin the post-pass. Recall that each $\boldsymbol{e}_{ik}^{RS} = \text{expsum}(\boldsymbol{a}_{ik}^{RS})$ and $\boldsymbol{u}_{ik}^{RS} = \text{softmax}(\boldsymbol{a}_{ik}^{RS})\boldsymbol{v}_k^S$. Combined, these give the same information as $\boldsymbol{e}_{ik}^{RS}$ and $\exp(\boldsymbol{a}_{ik}^{RS})\boldsymbol{v}_k^S$, as one can elementwise multiply $\boldsymbol{e}_{ik}^{RS}$ by $\boldsymbol{u}_{ik}^{RS}$ to get the latter, and elementwise divide the latter by $\boldsymbol{e}_{ik}^{RS}$ to get $\boldsymbol{u}_{ik}^{RS}$. Now, $\boldsymbol{e}_{ik}^{RS}$ is of shape $H \times |R_i|$, and obfuscates (by row summation) much of the information in the attention score submatrix $\boldsymbol{e}_{ik}^{RS}$ of shape $H \times |R_i| \times |S_k|$, assuming each $|S_k|$ is sufficiently large: so we focus on reversal from the $\beta$ shards $\exp(\boldsymbol{a}_{ik}^{RS})\boldsymbol{v}_k^S$ of shape $H \times |R_i| \times d$.

We consider this primarily in the context of the vocab-matching attack, though future work could certainly examine a learning reversal attack on this set of $\beta$ inputs. Indeed, since $\boldsymbol{a}_{ik}^{RS} = \boldsymbol{q}_i^R(\boldsymbol{k}_k^S)^T + \boldsymbol{m}_{ik}^{RS}$, assuming a unidirectional mask (vocab-matching does not apply if this is not true, anyways), we see that if $R_i$ contains elements $r_1 < r_2 < \ldots < r_k$, then the $l$th row of $\exp(\boldsymbol{a}_{ik}^{RS})\boldsymbol{v}_k^S$, which is known to CompNode$_i$, is exactly

$$\sum_{\substack{s < r_l \\ s \in S_k}} \exp\left(\boldsymbol{q}[:, r_l, :](\boldsymbol{k}[:, s, :])^T\right) \boldsymbol{v}[:, s, :].$$

At Layer 0, since $\boldsymbol{q}, \boldsymbol{k}, \boldsymbol{v}$ are linear projections of token embeddings, the above summation depends only on tokens in the set $\{r_l\} \cup \{s \in S_k : s < r_l\}$. Of these, CompNode$_i$ knows $r_l$ and all tokens in $R_i$, so the

unknown tokens in the summation are $\{s \in S_k : s < r_l\} \setminus R_i$. Thus, when we increase $l$ by 1, the set of *unknown* tokens this summation depends on will additionally include $\{s \in S_k : r_l < s < r_{l+1}\}$, i.e. the tokens in $S_k$ between token $r_l$ and $r_{l+1}$. If there are at most $\rho$ tokens in some gap, where $\rho$ is the vocab-matching threshold, then CompNode$_i$ can do a forward pass over all sequences of such $V^{\leq \rho}$ unknown tokens, until the summations match their known values. This is essentially the vocab-matching attack: to prevent it, we need each "unknown" gap $\{s \in S_k : r_l < s < r_{l+1}\}$ to have size $> \rho$. *It turns out that $(c, \delta)$-sharding for both $R$ and $S$ sharding satisfies this property: all such unknown gaps have size $c$, so for $c > \rho$, we prevent vocab-matching.*

## N    COST OPTIMIZATIONS

In Section 7.1, we gave a high-level overview of Cascade, and deferred discussions about optimization. Here, we discuss a few cost and communication optimizations, again assuming symmetry of $S$ and $T$ sharding.

**Caching**    After a new token is generated in Cascade, the CompNode holding that token will send it back to *one* of the existing CompNodes, and single-token generation will repeat to get the next token. To speed up generation after the first new token, the CompNodes and AttnNodes can store their partial intermediate states, and only the 1 CompNode and $\beta$ AttnNodes associated with the most recent token will need to participate in the single-token generation: this means KV-caching naturally extends to Cascade. Formally, suppose $n$ is the index of the most recently generated token, and it belongs to the hidden shard $R_i$ and the query shard $S_j$. Only CompNode$_i$ needs to perform new computation in generating the $(n+1)$st token, along with each AttnNode$_{jk}$ for $1 \leq k \leq \beta$: this is because only these AttnNodes require the $n$th query row. Furthermore, these $\beta + 1$ nodes, having stored intermediate results from previous forward passes, can avoid repeat computation of attention scores and earlier hidden states. Essentially, this results in token-sharded KV-caching.

**Symmetry Reduction**    We see that AttnNode$_{jk}$ and AttnNode$_{kj}$ actually have the exact same information in the worst-case: they both have access to indices $S_j \cup S_k$. Thus, at no loss of security, we can combine AttnNode$_{jk}$ and AttnNode$_{kj}$ into one node, thereby approximately halving the number of AttnNodes required, and reducing communication byte overhead.

**Synchronization**    A key assumption in our communication analysis from Appendix K was that nodes synchronize between stages. That is, AttnNodes wait until they all finish before sending information to CompNodes in parallel; and likewise for the CompNodes sending information to AttnNodes. But in practice, depending on the sharding scheme, synchronization is not necessary; and relaxing it can allow some nodes to proceed earlier than others. For instance, in a sharding scheme where CompNode$_1$ holds only the first $k$ tokens, because the first $k$ logits do not depend at all on tokens $k+1, \ldots, n$ in a unidirectional model, then CompNode$_1$ can proceed through all its forward passes without waiting for *any* information from other nodes. Future work could analyze the tradeoff between such synchronization relaxations, which are not possible with schemes like $(c, \delta)$-sharding, and token security.

## O    NOISING METHOD PERFORMANCE

Below, we provide exact (not only the maximum) ROUGE scores across layers $1, 6, 11, 16, 21, 26$, for all methods of noising discussed in Section 6. Table 13, Table 14, Table 15 show these results. We also provide a complete breakdown of LiveBench scores per category in Table 17.

Table 13: The ROUGE scores of decoded texts with added noise and no permutation.

| **Layer** | $\sigma = 10^{-2}$ | $\sigma = 10^{-1}$ | Random Emb | 8-bit quantization | 4-bit quantization |
|---|---|---|---|---|---|
| 1 | - | - | - | - | - |
| 6 | - | - | - | - | - |
| 11 | - | - | - | - | - |
| 16 | - | - | - | - | - |
| 21 | - | - | - | - | - |
| 26 | - | - | - | - | - |

Table 14: The ROUGE scores of decoded texts with added noise and sequence dimension permutation.

| Layer | $\sigma = 10^{-2}$ | $\sigma = 10^{-1}$ | Random Emb | 8-bit quantization | 4-bit quantization |
|-------|--------|--------|------------|--------------------|--------------------|
| 1  | 0.0736 | - | - | - | - |
| 6  | 0.0326 | - | - | - | - |
| 11 | 0.0306 | - | - | - | - |
| 16 | 0.0163 | - | - | - | - |
| 21 | 0.0184 | - | - | - | - |
| 26 | 0.0145 | - | - | - | - |

Table 15: The ROUGE scores of decoded texts with added noise and hidden dimension permutation.

| Layer | $\sigma = 10^{-2}$ | $\sigma = 10^{-1}$ | Random Emb | 8-bit quantization | 4-bit quantization |
|-------|--------|--------|------------|--------------------|--------------------|
| 1  | - | 0.0000 | - | - | - |
| 6  | - | 0.0000 | - | - | - |
| 11 | - | - | - | - | - |
| 16 | - | - | - | - | - |
| 21 | - | - | - | - | - |
| 26 | - | - | - | - | - |

Table 16: The ROUGE scores of decoded texts with added noise, sequence dimension permutation, and hidden dimension permutation.

| Layer | $\sigma = 10^{-2}$ | $\sigma = 10^{-1}$ | Random Emb | 8-bit quantization | 4-bit quantization |
|-------|--------|--------|------------|--------------------|--------------------|
| 1  | 0.0700 | 0.0000 | 0.1919 | - | 0.7050 |
| 6  | 0.0328 | 0.0000 | 0.0361 | - | 0.5779 |
| 11 | 0.0285 | 0.0012 | 0.0297 | - | 0.5722 |
| 16 | 0.0148 | 0.0017 | 0.0140 | - | 0.5651 |
| 21 | 0.0129 | 0.0030 | 0.0151 | - | 0.5568 |
| 26 | 0.0124 | 0.0104 | 0.0117 | - | 0.5564 |

Table 17: Performance of Gemma-2-2B-IT on LiveBench with added noise.

| Method | Avg. | Coding | Data Analysis | Instruction Following | Language | Math | Reasoning |
|--------|------|--------|---------------|-----------------------|----------|------|-----------|
| Baseline (no noise) | 20.7 | 9.4 | 26.1 | 48.9 | 15.2 | 13.1 | 11.3 |
| Gaussian, $\sigma = 10^{-2}$ | 21.0 | 11.1 | 27.4 | 51.2 | 13.7 | 13.4 | 9.3 |
| Gaussian, $\sigma = 10^{-1}$ | 1.2 | 0.0 | 0.0 | 6.9 | 0.4 | 0.0 | 0.0 |
| Random emb. prefix | 21.3 | 8.8 | 27.5 | 50.1 | 16.1 | 13.6 | 12.0 |
| 8-bit quantization | 20.2 | 8.8 | 27.1 | 49.2 | 13.3 | 13.0 | 10.0 |
| 4-bit quantization | 19.1 | 6.5 | 25.5 | 50.5 | 9.5 | 10.9 | 12.0 |

