# OpenReview forum: "Hidden No More: Attacking and Defending Private Third-Party LLM Inference"
_ICLR.cc/2025/Workshop/BuildingTrust — BuildingTrust_

### Official Review · Reviewer_SKau · 2025-02-16
**Simple yet effective prompt reconstruction attack**

**Rating:** 7
**Confidence:** 3

**Review:**

# Summary

- The paper proposes a very simple reconstruction technique that can almost completely recover the input prompt from hidden states.
- It demonstrates that the proposed method is robust against existing defenses such as Secure Multi-Party Computation (SMPC) and introduces a new sharding-based multi-party scheme called Cascade.

# Strength

- **Practicality**: The paper shows attention to practical details, such as implementing fuzzy matching to account for noise that may occur in computer systems. It is interesting that the introduced $\epsilon$ does not affect the attack success rate.
- **Extensive Experiments**: The method is shown to function not only in simple settings but also in cases where mechanisms like permutation, noise, and quantization act as defenses.
- **Defense**: The paper thoroughly organizes the features and drawbacks of existing defenses and proposes Cascade as a defense to neutralize the proposed attack. The analysis of Cascade (Section 7.2) is also extensive.

# Comments, Weakness

- **Intuitive Understanding**: Visual explanations or concrete examples of what is happening during reconstruction might aid in understanding.
- **Runtime**: Table 4 presents the measurement results using BERT, but can it scale to more recent, larger models (or multilingual models with a larger vocabulary)? Although 4.2.2 mentions one example (it takes < 30 seconds to reconstruct 50 token prompts), readers would benefit from rigorous and extensive experiments in this direction (e.g., the correlation between runtime and model size / target layer depth / vocab size / etc).

Overall, the proposal is simple yet effective, with significant practical implications. If, as the authors claim, no similar attacks exist, this paper represents a substantial contribution.

---

### Official Review · Reviewer_zvkW · 2025-02-25
**Interesting topic, presentation could be improved.**

**Rating:** 6
**Confidence:** 3

**Review:**

In general, I find the addressed topic and the paper interesting and could be a good addition to the workshop. However, I have certain concerns that I strongly advise the authors to address before they make any copy of the manuscript public:

- The basic ideas behind the attack presented in Sections 4 and 5 are in principle similar to the attack in [1], while [1] deals with the more difficult case of gradient inversion. While I believe that the ideas can be independently discovered, as they are relatively simple, relation to prior work still has to be discussed.
- In general, a stronger connection to gradient inversion and model inversion literature should be established. I understand that this is an inference-time attack, while those are trianing-time attacks, certain techniques, as seen above, translate between these attack types.
- The significance of the $\epsilon$ threshold in the matching procedure is unclear. I wonder why simply taking the min L1 distance would not be sufficient. This is anyway what the algorithm falls back to if no matches within $\epsilon$ distance are found. Also, if the $\epsilon$ match is unique, it is then necessarily the min.
- Regarding the proposed defense; if I understood it correctly, there is still a node that has access to all logits in the end. In this way, it might be possible to apply a similar auto-regressive enumaration attack as the one presented in Sections 4 and 5, but instead of looking at the hidden state, looking at the logits. The attacker here could be the owner of this node, and as we assume that the attacker has access to the full weights of the model, they could run the model in shadow to match the logits.

Some higher-level comments:
- In general, I find the structuring of the paper rather confusing. Method-related and experimental sections are intermixed, and experimental results are embedded into technique presentations. The paper would benefit from a clear separation of technical and experimental sections. The experiments themselves would also benefit from a more extensive introduction, clearly setting up the experimental paramaters.
- The threat model has to be also clarified better. It is unclear what role the fact plays that the model is open-source. Naturally, any hosting service will have access to the model weights, no matter if those are publicly accessable or not. It is also unclear why such hosting services would be willingly giving up parts of their inference infrastructure and share inference it with other parties. Also, why could these parties not collude and exchange information about the hidden states on their nodes? Finally, it would also help the clarity of the threat model if the authors would discuss related hardware-level confidentiality techniques, i.e., trusted execution environments.

---

### Official Review · Reviewer_rJck · 2025-02-28
**This paper reveals a potent new attack on hidden-state privacy and introduces a scalable multi-party token-sharding defense, Cascade, that resists existing reversal techniques and offers a practical alternative to heavy MPC-based solutions. It highlights the urgent need for robust privacy mechanisms in open-weights LLM inference.**

**Rating:** 9
**Confidence:** 5

**Review:**

This paper deserves an extensive review so here we go :

Quality:
This paper provides a thorough investigation of privacy vulnerabilities in third-party LLM inference under the open-weights setting. The authors introduce a simple yet powerful “vocab-matching” attack capable of recovering original user prompts with near-perfect accuracy, even when hidden states are permuted or injected with noise. Through extensive experiments on different state-of-the-art LLM architectures (Gemma-2-2B-IT, Llama-3.1-8B-Instruct) and hidden-layer settings, the paper demonstrates a broad range of successful attacks that defeat several existing defenses. The proposed defense, Cascade, is a token-sharding multi-party inference scheme, which trades off certain cryptographic guarantees in order to achieve significantly faster and more communication-efficient privacy-preserving inference. The paper’s security analysis includes both brute-force style and learning-based attacks, and the authors provide detailed cost (computation and communication) analyses relative to alternative MPC approaches. Overall, the methodology is well-structured, and the empirical evaluations are convincing.

Clarity:
The paper is clear in its exposition of the attack setup, threat model, and the design of Cascade. The theoretical details behind the vocab-matching attack and its variants (to handle permuted or noised hidden states) are spelled out systematically, and the step-by-step algorithms make the approach and experimental procedures quite transparent. Cascade’s architecture, while conceptually non-trivial, is broken down into discrete multi-party steps—pre-pass, attention-pass, and post-pass—making it easier to follow how the token-level partitioning is actually performed. A few points (e.g., real-world latency of Cascade, hardware differences in floating-point arithmetic) might benefit from additional elaboration, but these do not detract significantly from the overall clarity.

Originality:
The paper’s central contribution is a new line of attack—an efficient sequential vocab-matching procedure—specifically tailored to exploit the autoregressive structure of modern LLMs in an open-weights environment. While past work has explored embedding inversion and hidden-state reconstruction, few have shown near-perfect text recovery against sophisticated defenses such as permutation-based schemes and noise injection.
The proposed Cascade protocol, leveraging token-wise partitioning across multiple untrusted nodes, is also novel in how it avoids the heavy overhead typical of full-fledged MPC.

Significance:
1. Attack Implications: The demonstration that hidden-state permutations or moderate noise do not suffice to hide user inputs underscores a critical vulnerability in certain existing “lightweight” privacy solutions. This is directly relevant to the workshop’s focus on trust and safety in LLM applications, particularly in regulated domains (e.g. healthcare).
2. Defensive Contribution: Cascade offers a meaningful middle ground on the privacy–efficiency spectrum, which practitioners may find more practical than full-blown cryptographic protocols. The authors show strong empirical gains (over 100× faster than prior MPC methods on Bert-sized models), suggesting wide potential impact if integrated into real-world distributed inference pipelines.

Pros
1. High Attack Success: Vocab-matching recovers user prompts with remarkable accuracy under many transformations (permutation, noise, quantization) at multiple layers of large modern LLMs.
2. Detailed Evaluation: Exhaustive experiments cover permutations across different dimensions (sequence, hidden, factorized 2D) and various noise/quantization schemes, providing convincing evidence of the vulnerability.
3. Novel Defense: Cascade’s use of token sharding is presented as an efficient multi-party scheme that counters both brute-force and learning-based inversion attacks.
4. Extensive Analysis: The paper includes thorough theoretical discussion, security analysis, and comparisons with prior cryptographic protocols (quantitative metrics for runtime and communication).

Cons
1. Layer 0 Limitation: The paper concedes that if layer-0 embeddings are exposed in any form, token reconstruction is trivial. Cascade thus only achieves security from layer 1 onward, restricting its immediate deployment for fully securing all tokens.
2. No Formal Cryptographic Guarantee: Cascade’s defense is described in terms of sharding’s statistical obfuscation rather than a strict cryptographic proof. While this design choice is deliberate, it may leave room for stronger adaptive or combined attacks in the future.
3. Scalability Nuances: Real-world systems might see performance hits from high latency or unreliable nodes, and the paper’s evaluation largely assumes ideal parallel transport and bandwidth, warranting more real-network benchmarks.
4. Complex Parameter Tuning: Security depends on the choice of shard sizes (c, δ, α, β), which is left somewhat open-ended. This could pose an adoption barrier for users unfamiliar with multi-node distribution strategies.

---

### Decision · Program_Chairs · 2025-03-04

Accept